# Towards Resilient Safety-driven Unlearning for Diffusion Models against Downstream Fine-tuning

**Boheng Li[1], Renjie Gu[2], Junjie Wang[3], Leyi Qi[4], Yiming Li[1]***
**Run Wang[3], Zhan Qin[4], Tianwei Zhang[1]**
[1]Nanyang Technological University, Singapore    [2]Central South University, China
[3]Key Laboratory of Aerospace Information Security and Trusted Computing,
Ministry of Education, School of Cyber Science and Engineering, Wuhan University, China
[4]State Key Laboratory of Blockchain and Data Security, Zhejiang University, China

## Abstract

Text-to-image (T2I) diffusion models have achieved impressive image generation quality and are increasingly fine-tuned for personalized applications. However, these models often inherit unsafe behaviors from toxic pretraining data, raising growing safety concerns. While recent safety-driven unlearning methods have made promising progress in suppressing model toxicity, they are found to be fragile to downstream fine-tuning, as we reveal that state-of-the-art methods largely fail to retain their effectiveness even when fine-tuned on entirely benign datasets. To mitigate this problem, in this paper, we propose ResAlign, a safety-driven unlearning framework with enhanced resilience against downstream fine-tuning. By modeling downstream fine-tuning as an implicit optimization problem with a Moreau envelope-based reformulation, ResAlign enables efficient gradient estimation to minimize the recovery of harmful behaviors. Additionally, a meta-learning strategy is proposed to simulate a diverse distribution of fine-tuning scenarios to improve generalization. Extensive experiments across a wide range of datasets, fine-tuning methods, and configurations demonstrate that ResAlign consistently outperforms prior unlearning approaches in retaining safety, while effectively preserving benign generation capability. Our code and pretrained models are publicly available here.

⚠️ *Disclaimer: This paper includes AI-generated images containing partially nude human figures and other sensitive content, shown only for research purposes.*

## 1 Introduction

Text-to-image (T2I) diffusion models have emerged as a dominant class of generative AI due to their unprecedented ability to synthesize high-quality, diverse, and aesthetically compelling general images from natural language descriptions [67, 58]. Beyond synthesizing general images, there is also a growing interest in customizing pretrained models for personalized generation, e.g., generating images of specific facial identities or artistic styles that are underrepresented in the original training data [45]. This is typically achieved by fine-tuning the pretrained *base model* on a small reference dataset for a few steps [69, 39, 76]. The development of several advanced fine-tuning methods [24, 69, 20] as well as the rapid proliferation of "fine-tuning-as-a-service" platforms [5] has made personalization widely accessible, fueling a surge in applications such as stylized avatars, fan art, and thematic illustrations, which are becoming increasingly popular especially among younger users [74].

Yet alongside the rapid advancements of diffusion models, growing concerns have emerged regarding their potential to generate inappropriate or harmful content (e.g., sexually explicit imagery) [13, 70].

---

*Corresponding Author: Yiming Li (e-mail: `liyiming.tech@gmail.com`).

39th Conference on Neural Information Processing Systems (NeurIPS 2025).

Due to the large-scale and web-crawled nature of their training datasets, modern T2I models inevitably ingest amounts of harmful material during pretraining [71]. As a result, these models can reproduce such content either when explicitly prompted or inadvertently triggered. For example, recent studies [70, 61, 40] based on real-world user generations demonstrate that widely-deployed models like Stable Diffusion [67] are particularly prone to producing unsafe content, even though many of the prompts that lead to unsafe outputs appear benign and may not be intended to generate harmful results. These vulnerabilities not only allow malicious exploitation through direct or adversarial prompting [79, 84], but also increase the risk of harmful unintended exposure for ordinary, benign users [40], raising serious ethical concerns for real-world deployment. In response to these concerns, a variety of safety-driven unlearning methods [13, 43, 83] have recently been proposed to modify the pretrained models' parameters in order to suppress their capacity for unsafe generation.

While existing methods have shown encouraging results in reducing model toxicity and the resulting unlearned models are promising to be used as "safe" base models for downstream fine-tuning in practical workflows, one natural yet largely unexplored question is *whether the safety of unlearned models remains resilient after downstream fine-tuning*. Unfortunately, recent studies [56, 15] have shown that many existing methods can be easily reversed, where fine-tuning on *harmful* samples for as few as 20 steps can largely recover a model's unsafe capability. More strikingly, our empirical results reveal that even when fine-tuned on purely *benign* data, state-of-the-art unlearning methods can regress, with the model's harmfulness approaching its pre-unlearning state. In other words, even entirely benign users without any malicious intent or harmful data may inadvertently trigger a recovery of unsafe behaviors, posing unforeseen safety risks in real-world use. These findings suggest that current methods may be significantly more brittle than previously assumed and are largely unprepared to serve as reliably safe base models for downstream fine-tuning, underscoring the urgent need for more resilient approaches that can withstand post-unlearning adaptation.

Towards this end, in this paper, we propose a resilient safety-driven unlearning framework dubbed ResAlign to mitigate the aforementioned problem. The intuition behind our method is that *unlearning should not only suppress harmfulness at the current model state, but also explicitly minimize the degree to which harmful behaviors can be regained after (simulated) downstream fine-tuning*. While conceptually simple, it is particularly challenging to develop a principled and efficient optimization framework to realize this objective. This is because fine-tuning itself is a multi-step optimization process, making it non-trivial to predict which update direction on the original parameters helps minimize the regained harmfulness after downstream fine-tuning. To address this, we approximate fine-tuning as an implicit optimization problem with a Moreau envelope formulation [52, 63], which enables efficient gradient estimation via implicit differentiation. Besides, to ensure generalizability against the wide variability in real-world downstream fine-tuning procedures (e.g., different datasets, fine-tuning methods, and hyperparameters), we design a meta-learning approach that simulates a distribution of plausible fine-tuning configurations during training, allowing the model to generalize its resilience across a broad range of downstream adaptation scenarios. We also provide insights from the theoretical perspective to explain the empirical effectiveness of our method.

In conclusion, our main contributions are threefold. **(1)** We empirically reveal that existing safety-driven unlearning methods largely fail to retain their effectiveness after downstream fine-tuning, even when the data does not contain unsafe content. **(2)** We propose ResAlign, a resilient safety-driven unlearning framework for T2I diffusion models. By leveraging a Moreau envelope-based approximation and a meta-learning strategy over diverse adaptation scenarios, ResAlign explicitly accounts for and minimizes post-unlearning degradation due to downstream fine-tuning efficiently with high generalizability. We further provide theoretical insights to help understand the empirical effectiveness of our method. **(3)** Through extensive experiments, we show that ResAlign consistently outperforms baselines in maintaining safety after fine-tuning, and generalizes well to a wide range of advanced fine-tuning methods, datasets, and hyperparameters. It also preserves both general and personalized generation quality well, generalizes across various diffusion models and loss functions, and remains effective even under harmful data contamination or adaptive attacks.

## 2 Background & Related Work

**Text-to-Image (T2I) Diffusion Models.** Diffusion models are probabilistic generative models that learn the data distribution by reversing a predefined forward noising process through iterative denoising [22, 67]. Given a clean sample $x$ and a predefined noise schedule $\{\alpha_t, \sigma_t\}_{t=1}^T$, the forward

diffusion process constructs a noisy version at timestep $t$ as $x_t = \alpha_t x + \sigma_t \epsilon$, where $\epsilon \sim \mathcal{N}(0, I)$. Then, a denoising neural network $\hat{\epsilon}_\theta$ parameterized by $\theta$ is trained to predict the added noise $\epsilon$ from $x_t$ and the timestep $t$. In the context of text-to-image generation [67], the prediction is further conditioned on a natural language prompt $p$. The model thus learns a conditional denoising function $\hat{\epsilon}_\theta(x_t, p, t)$, and its training objective is to minimize the following denoising score matching loss:

$$\mathcal{L}_{\text{denoise}}(\theta, \mathcal{D}_{\text{train}}) = \mathbb{E}_{\epsilon, t, (x, p) \sim \mathcal{D}_{\text{train}}} \left[ w_t \cdot \| \hat{\epsilon}_\theta(\alpha_t x + \sigma_t \epsilon, p, t) - \epsilon \|_2^2 \right], \tag{1}$$

where $x$ is the ground-truth image, $p$ is the text condition, $t$ is uniformly sampled from $\{1, \ldots, T\}$, and $w_t$ is a weighting factor that balances noise levels. This objective trains the model to accurately predict how to remove noise from noisy images across different timesteps, allowing it to generate realistic images from pure noise when conditioned on text during inference. Currently, the Stable Diffusion series [67, 58], which operate the diffusion process in a learned latent space and are pretrained on large-scale datasets, stands as the most widely adopted open-source model for T2I generation. They are also increasingly fine-tuned on downstream datasets to learn and generate concepts that are absent in their pretraining datasets for customized generation purposes [69, 57, 36].

**Mitigating Unsafe Generation in T2I Models.** To mitigate unsafe generation in T2I models, several strategies have been developed recently [60, 78, 48, 13], which can be broadly categorized into *detection-based* and *unlearning-based*. Detection-based strategies deploy external safety filters (e.g., DNN-based harmful prompt/image detectors) to inspect and block unsafe requests [7, 48]. While effective, they do not inherently "detoxify" the diffusion model, often introducing non-negligible additional computational and memory overhead during inference [43], and can be easily removed once the model is open-sourced [64]. On the other hand, unlearning-based approaches internalize safety constraints by directly modifying model parameters to reduce unsafe generation at its source [13, 35, 43]. At a high level, given a pretrained T2I diffusion model $\theta \in \mathbb{R}^d$, existing safety-driven unlearning approaches typically aim to optimize the model with the following objective:

$$\theta^* \in \arg\min_{\theta \in \Theta} \mathcal{L}_{\text{harmful}}(\theta) + \alpha \mathcal{R}(\theta), \tag{2}$$

where $\mathcal{L}_{\text{harmful}}(\theta)$ is a harmful loss such that minimizing it encourages the model to unlearn harmful behaviors, and $\mathcal{R}(\theta)$ is a utility-preserving regularization term that maintains the model's benign generation capabilities. The hyperparameter $\alpha > 0$ is a weighting coefficient. Since the seminal work of CA [35] and ESD [13], safety-driven unlearning has rapidly gained traction, with a growing line of subsequent works expanding the design space across multiple dimensions including loss function design [35, 13], sets of updated parameters [43, 13], optimization strategies [14, 83, 18], and others.

Despite this progress, a recent pioneering work [56] has identified a malicious fine-tuning issue, where unlearned models can quickly regain harmful generative abilities after fine-tuning on a small set of *unsafe* images, producing diverse outputs that resemble the quality and variety of the original pre-unlearned model. To the best of our knowledge, the most advanced attempt to address this issue is LCFDSD [56], which encourages separation in the latent space between clean and harmful data distributions to increase the difficulty of recovering unsafe behaviors. However, this comes with a notable degradation in the benign utility of the model and only brings slight improvements over existing unlearning methods, as validated in our experiments (Sec. 4.2). To summarize, how to effectively mitigate the post-fine-tuning resurgence of harmful capabilities in unlearned models remains an important yet largely unaddressed problem and is worth further exploration.

## 3  Methodology

**Motivation & Problem Formulation.** Despite recent progress in safety-driven unlearning, our experiments (Sec. 4.2) reveal that many state-of-the-art methods fail to retain their effectiveness after downstream fine-tuning, *even when the adaptation data is entirely benign* (instead of solely with *harmful* data). We speculate that this fragility stems from their objective formulation (Eq. (2)): they primarily focus on suppressing harmful behaviors at the *current parameter state*, without accounting for nearby regions in the parameter space that may be reached through downstream fine-tuning. As a result, the local neighborhood of the unlearned model may still be vulnerable, such that even benign gradient signals during fine-tuning can inadvertently push the model into toxic regions and erode its safety. Motivated by this observation, we aim to learn model parameters that are not only safe in their current state, but also lie in regions of the parameter space where safety is preserved under downstream

fine-tuning. In other words, we seek to explicitly *penalize the increase in harmfulness loss caused by (simulated) downstream fine-tuning during unlearning*. Formally, given a pretrained T2I diffusion model parameterized by $\theta \in \mathbb{R}^d$, we propose the following resilient unlearning objective:

$$\theta^* \in \arg\min_{\theta \in \Theta} \ \mathcal{L}_{\text{harmful}}(\theta) + \alpha\mathcal{R}(\theta) + \beta\big[\mathcal{L}_{\text{harmful}}(\theta_{\text{FT}}^*) - \mathcal{L}_{\text{harmful}}(\theta)\big], \tag{3}$$

where $\theta_{\text{FT}}^* = \text{ADAPT}(\theta, \mathcal{D}_{\text{FT}}, \mathcal{C})$ represents the parameters obtained by adapting $\theta$ on dataset $\mathcal{D}_{\text{FT}}$ with configuration $\mathcal{C}$. For example, if $\mathcal{C}$ is the standard gradient descent for $T$ steps with a learning rate of $\eta$, then we have $\theta_{\text{FT}}^* = \theta - \eta\sum_{t=0}^{T-1}\nabla_{\theta^{(t)}}\mathcal{L}_{\text{FT}}(\theta^{(t)}, \mathcal{D}_{\text{FT}})$, where $\mathcal{L}_{\text{FT}}$ represents the standard diffusion denoising loss and $\theta^{(i)}$ indicates the model parameters at the $i$-th step with $\theta^{(0)} = \theta$. $\alpha$ and $\beta$ are hyperparameters that balance the three terms. In this paper, we follow previous works [85, 83] and instantiate $\mathcal{L}_{\text{harmful}}(\theta)$ as the negative denoising score matching loss on a set of inappropriate prompt-image pairs and $\mathcal{R}(\theta)$ as the distillation loss that minimizes the difference of noise prediction between the current model and the original model $\hat{\epsilon}_{\theta_0}$ on a set of preservation prompts, i.e., $\mathcal{L}_{\text{harmful}}(\theta) = -\mathbb{E}_{\epsilon,t,(x,p)\sim\mathcal{D}_{\text{harmful}}}[w_t \cdot \|\hat{\epsilon}_\theta(\alpha_t x + \sigma_t\epsilon, p, t) - \epsilon\|_2^2]$ and $\mathcal{R}(\theta) = \mathbb{E}_{\epsilon,t,(x_0,p)\sim\mathcal{D}_{\text{preserve}}}[w_t \cdot \|\hat{\epsilon}_\theta(x_t, p, t) - \hat{\epsilon}_{\theta_0}(x_t, p, t)\|_2^2]$. Note that our method is also compatible with other designs of these losses, as long as they offer similar utility (see results in Sec. 5).

**Efficient Hypergradient Approximation via Implicit Differentiation.** To optimize the resilient unlearning objective in Eq. (3), one natural idea is to use gradient-based optimization. Thus, we take gradients with respect to $\theta$ and rearrange terms. This yields the following expression:

$$(1-\beta)\nabla_\theta\mathcal{L}_{\text{harmful}}(\theta) + \alpha\nabla_\theta\mathcal{R}(\theta) + \beta\nabla_\theta\mathcal{L}_{\text{harmful}}(\theta_{\text{FT}}^*). \tag{4}$$

The first two terms in Eq. (4) are standard and can be easily computed via backpropagation. The difficulty lies in the last term, i.e., the *hypergradient* $\nabla_\theta\mathcal{L}_{\text{harmful}}(\theta_{\text{FT}}^*)$. Because $\theta_{\text{FT}}^*$ is an implicit function of $\theta$, according to the chain rule, we have $\nabla_\theta\mathcal{L}_{\text{harmful}}(\theta_{\text{FT}}^*) = (\partial\theta_{\text{FT}}^*/\partial\theta)^\top \cdot \nabla_{\theta_{\text{FT}}^*}\mathcal{L}_{\text{harmful}}(\theta_{\text{FT}}^*)$, which involves the Jacobian $\partial\theta_{\text{FT}}^*/\partial\theta$. Since $\theta_{\text{FT}}^*$ is typically obtained through $T$-step iterative gradient descent, this derivative expands into a product of Jacobians $\partial\theta_{\text{FT}}^*/\partial\theta = \prod_{t=0}^{T-1}\left(I - \eta\nabla_{\theta^{(t)}}^2\mathcal{L}_{\text{FT}}(\theta^{(t)}, \mathcal{D}_{\text{FT}})\right)$, where $\nabla_{\theta^{(t)}}^2\mathcal{L}_{\text{FT}}(\theta^{(t)}, \mathcal{D}_{\text{FT}})$ is the Hessian matrix of the fine-tuning loss w.r.t. $\theta^{(t)}$. As such, direct computation requires storing and backpropagating through the entire trajectory and their Hessian matrices, which is computationally and memory-prohibitive.

To address this issue, inspired by recent work in bilevel optimization [47, 16, 81, 63], we approximate fine-tuning as an implicit optimization over the Moreau envelope [52] of the loss. Specifically, we approximately regard the solution of downstream fine-tuning (i.e., the resulting parameters $\theta_{\text{FT}}^*$) as the minimizer of the following Moreau envelope (ME) [52, 16, 63], i.e.,

$$\theta_{\text{FT}}^* \in \arg\min_{\theta'} \ \mathcal{L}_{\text{FT}}(\theta', \mathcal{D}_{\text{FT}}) + \frac{1}{2\gamma}\|\theta' - \theta\|^2, \tag{5}$$

where $\gamma$ (set as 1 in this paper) is a proximity coefficient [63]. Intuitively, when fine-tuning diffusion models, we typically start from a strong pretrained base model and only take a few gradient steps. This implicitly restricts the solution $\theta_{\text{FT}}^*$ to remain near $\theta$, making it well-approximated by the minimizer of the proximal objective in Eq. (5). As a result, $\theta_{\text{FT}}^*$ naturally satisfies the first-order optimality condition, i.e., $\nabla_{\theta_{\text{FT}}^*(\theta)}\mathcal{L}_{\text{FT}}(\theta_{\text{FT}}^*, \mathcal{D}_{\text{FT}}) + \frac{1}{\gamma}(\theta_{\text{FT}}^* - \theta) = 0$ according to the Karush-Kuhn-Tucker theorem [34, 63]. Since $\theta_{\text{FT}}^*$ is implicitly defined as a function of $\theta$ through this optimality condition, we apply the implicit function theorem to differentiate both sides with respect to $\theta$. Then, by right-multiplying both sides by $\nabla_{\theta_{\text{FT}}^*}\mathcal{L}_{\text{harmful}}(\theta_{\text{FT}}^*)$, we can derive the following equation (details in Appendix A):

$$\left(\nabla_{\theta_{\text{FT}}^*}^2\mathcal{L}_{\text{FT}}(\theta_{\text{FT}}^*, \mathcal{D}_{\text{FT}}) + \frac{1}{\gamma}I\right) \cdot \nabla_\theta\mathcal{L}_{\text{harmful}}(\theta_{\text{FT}}^*) = \frac{1}{\gamma}\nabla_{\theta_{\text{FT}}^*}\mathcal{L}_{\text{harmful}}(\theta_{\text{FT}}^*) \tag{6}$$

Let $A := \nabla_{\theta_{\text{FT}}^*}^2\mathcal{L}_{\text{FT}} + \frac{1}{\gamma}I$, $x := \nabla_\theta\mathcal{L}_{\text{harmful}}(\theta_{\text{FT}}^*)$, and $b := \frac{1}{\gamma}\nabla_{\theta_{\text{FT}}^*}\mathcal{L}_{\text{harmful}}$, we observe from Eq. (6) that $\nabla_\theta\mathcal{L}_{\text{harmful}}(\theta_{\text{FT}}^*)$ is essentially the solution to the linear system $Ax = b$, which can be efficiently solved using the Richardson iteration method [66] (setting the relaxation parameter as $\omega = \gamma = 1$):

$$x^{(k+1)} = \gamma b - \gamma\nabla_{\theta_{\text{FT}}^*}^2\mathcal{L}_{\text{FT}}(\theta_{\text{FT}}^*, \mathcal{D}_{\text{FT}}) \cdot x^{(k)}, \tag{7}$$

with initialization $x^{(0)} = 0$. Lastly, we take the final iterate $x^{(K)}$ as the implicit gradient $\nabla_\theta\mathcal{L}_{\text{harmful}}(\theta_{\text{FT}}^*)$ and plug it into Eq. (3), which enables end-to-end gradient-based optimization.

Our approach has the following advantages. First, computing the implicit gradient $\nabla_\theta \mathcal{L}_{\text{harmful}}(\theta_{\text{FT}}^*)$ only depends on the final fine-tuned parameters $\theta_{\text{FT}}^*$ and the local Hessian-vector product $\nabla_{\theta_{\text{FT}}}^2 \mathcal{L}_{\text{FT}} \cdot x$. This product can be efficiently computed as a Hessian-vector product (HVP), which is readily supported in modern autodiff frameworks such as PyTorch (via backward-mode automatic differentiation),

without explicitly forming or inverting any second-order Hessian matrices. Besides, as the gradient is entirely determined by $\theta_{\text{FT}}^*$, there is no need to store the intermediate models during fine-tuning, which significantly reduces memory overhead. Finally, we observe that the Richardson iteration used to approximate the implicit gradient (i.e., Eq. (7)) converges quickly within a few steps (5 in this paper), enabling efficient training for large-scale diffusion models. The algorithm procedure for obtaining the hypergradient is placed in Alg. 1.

---

**Algorithm 1** GETHYPERGRAD

1: **Input:** Base parameters $\theta$, downstream fine-tuned model $\theta_{\text{FT}}^*$, loss functions $\mathcal{L}_{\text{FT}}, \mathcal{L}_{\text{harmful}}$, proximity coefficient $\gamma$, number of steps $K$
2: **Output:** Hypergradient $\nabla_\theta \mathcal{L}_{\text{harmful}}(\theta_{\text{FT}}^*)$
3: $x^{(0)} \leftarrow 0$, $b \leftarrow \frac{1}{\gamma} \nabla_{\theta_{\text{FT}}} \mathcal{L}_{\text{harmful}}(\theta_{\text{FT}})$
4: **for** $k = 0$ to $K - 1$ **do**
5:     Compute Hessian-vector product $\nabla_{\theta_{\text{FT}}}^2 \mathcal{L}_{\text{FT}}(\theta_{\text{FT}}) \cdot x^{(k)}$ via iterative solver and reverse-mode autodiff
6:     $x^{(k+1)} \leftarrow \gamma b - \gamma \nabla_{\theta_{\text{FT}}}^2 \mathcal{L}_{\text{FT}}(\theta_{\text{FT}}) \cdot x^{(k)}$
7: **end for**
8: **Return** $x^{(K)}$

---

**Cross-configuration Generalization via Meta Learning.** So far, we have successfully derived a trajectory-independent hypergradient based on a Moreau envelope approximation, which can be efficiently computed as long as $\theta_{\text{FT}}^*$ is available. However, this requires access to the downstream fine-tuning configuration $\mathcal{C}$ and dataset $\mathcal{D}_{\text{FT}}$ during unlearning time, which is typically infeasible. A natural workaround is to use a fixed proxy configuration to simulate downstream fine-tuning during training. Yet, such models carry a risk of overfitting to the fixed simulated configuration [49], which may hinder generalization to other settings, especially those unseen during training.

To this end, we draw inspiration from meta-learning [10, 63, 53] to improve the cross-configuration generalization of ResAlign. Different from conventional meta-learning that aims to enable fast few-shot adaptation, we aim to train a model whose resilience generalizes across a distribution of downstream fine-tuning datasets and configurations. Specifically, we treat the fine-tuning configuration $\mathcal{C}$ (including loss function, optimizer, learning rate, and number of steps) and data sampling $\mathcal{D}_{\text{FT}}$ as meta-variables. During each inner loop iteration, we sample a batch of data $\mathcal{D}_{\text{FT}} \sim \mathcal{D}$ and configuration

---

**Algorithm 2** ResAlign

1: **Input:** Initial model parameters $\theta_0$, number of outer loop iterations $I$, number of inner loop iterations $J$, dataset $\mathcal{D}$, distribution over configurations $\pi(\mathcal{C})$, loss functions $\mathcal{L}_{\text{FT}}, \mathcal{L}_{\text{harmful}}, \mathcal{R}$, learning rate $\eta$, hyperparameters $\alpha, \beta, \gamma, K$
2: **Output:** Final unlearned model parameters $\theta_I$
3: **for** $i = 0$ to $I - 1$ **do**         ▷ Outer Loop
4:     $g_i \leftarrow (1 - \beta)\nabla_{\theta_i} \mathcal{L}_{\text{harmful}}(\theta_i) + \alpha \nabla_{\theta_i} \mathcal{R}(\theta_i)$
5:     **for** $j = 1$ to $J$ **do**         ▷ Inner Loop
6:         Sample data $\mathcal{D}_{\text{FT}} \sim \mathcal{D}$ and configuration $\mathcal{C} \sim \pi(\mathcal{C})$
7:         $\theta_{\text{FT}_{i,j}}^* \leftarrow \text{ADAPT}(\theta_i, \mathcal{D}_{\text{FT}}, \mathcal{C})$
8:         $g_{i,j} \leftarrow \text{GETHYPERGRAD}(\theta_i, \theta_{\text{FT}_{i,j}}^*, \mathcal{L}_{\text{FT}}, \mathcal{L}_{\text{harmful}}, \gamma, K)$
9:     **end for**
10:     $g_i \leftarrow g_i + \frac{\beta}{J} \sum_j^J g_{i,j}$     ▷ Aggregate hypergradients
11:     $\theta_{i+1} \leftarrow \theta_i - \eta \cdot g_i$
12: **end for**
13: **Return** $\theta_I$

---

$\mathcal{C} \sim \pi(\mathcal{C})$ from the pool of these meta-variable choices. Then, we run an adaptation process to obtain the fine-tuned model $\theta_{\text{FT}}^* = \text{ADAPT}(\theta, \mathcal{D}_{\text{FT}}, \mathcal{C})$ and use Alg. 1 to compute a hypergradient with respect to $\theta$. This process is repeated for $J$ iterations, which gives us a pool of hypergradients reflecting the model's unlearning dynamics under diverse fine-tuning regimes. Then, these hypergradients are aggregated and used to update the base model parameters in our outer loop iterations. Following previous works [10, 49], a first-order approximation is used for efficiency. Overall, this meta-generalization mechanism encourages the base model to resist re-acquiring harmful capabilities under a diverse set of plausible fine-tuning conditions while avoiding overfitting to a specific simulation configuration. The overall algorithm procedure is summarized in Alg. 2.

**Theoretical Insights.** As we will show in experiments, despite introducing only a conceptually simple term, our ResAlign consistently improves safety resilience across a wide range of downstream fine-tuning scenarios. To better understand the (potential) underlying mechanism behind ResAlign's empirical effectiveness, in this section, we provide a qualitative theoretical analysis, as follows:

**Proposition 1.** *Let $\theta \in \mathbb{R}^d$ and $\theta_{FT}^* \in \mathbb{R}^d$ denote the parameters of the base model and the fine-tuned model, respectively. Assume the harmful loss $\mathcal{L}_{harmful}$ is twice differentiable around $\theta$, the parameter*

*difference of the two models $\xi \in \mathbb{R}^d$ is small and can be regarded as a scaled version of a unit random variable $z \in \mathbb{R}^d$ with scaling factor $\sigma \in \mathbb{R}$, i.e., $\xi = \sigma z$. If we exploit the second-order Taylor expansion around model parameters and ignore higher-order terms, we have:*

$$\arg\min_{\theta} \ \mathbb{E}\big[\mathcal{L}_{harmful}(\theta^*_{FT}) - \mathcal{L}_{harmful}(\theta)\big] \approx \arg\min_{\theta} \ Tr(\nabla^2_\theta \mathcal{L}_{harmful}(\theta)), \qquad (8)$$

*where $Tr(\nabla^2_\theta \mathcal{L}_{harmful}(\theta)) = \sum_{i=1}^d \frac{\partial^2 \mathcal{L}_{harmful}(\theta)}{\partial \theta_i^2}$ is trace of the Hessian matrix of $\mathcal{L}_{harmful}$ w.r.t. $\theta$.*

In general, Proposition 1 shows that our additional term can be regarded as an implicit penalty on the trace of the Hessian of the harmfulness loss with respect to $\theta$. The trace of the Hessian is a well-known indicator of the overall curvature of the loss landscape [29, 11], where larger values reflect sharper minima that are highly sensitive to parameter perturbations, while smaller values correspond to flatter and more stable regions. Interestingly, prior works in generalization and robustness [32, 11] have shown that SGD-style optimizers tend to lead models to sharp minima, i.e., regions in the loss landscape with high curvature that are sensitive to parameter perturbations. This connection offers a potential explanation for the fragility of existing unlearning methods: since they do not explicitly regularize the curvature of the harmfulness loss, the resulting models, while appearing safe at their current parameter state, may lie in locally sharp and unstable regions of the loss landscape. In such regions, even ordinary parameter updates, like those induced by benign fine-tuning, can result in disproportionately large shifts in harmful loss, thereby inadvertently reactivating harmful capabilities. In contrast, our ResAlign introduces an implicit penalty on the trace of the Hessian of the harmfulness loss, encouraging convergence to flatter regions of the loss surface. This may help reduce the model's sensitivity to downstream updates and thereby improves its resilience to post-unlearning fine-tuning. We provide a proof of Proposition 1 and further discussion in Appendix A.

## 4 Experiments

### 4.1 Experimental Setup

**Baselines & Diffusion Models.** We compare our method with 6 text-to-image diffusion models, including SD v1.4 [67], ESD [13], SafeGen [43], AdvUnlearn [83], and two variants of LCFDSD [56] (i.e., LCFDSD-NG and LCFDSD-LT). We directly use their provided checkpoints or run their official code with default hyperparameters to obtain their unlearned models (see more details in Appendix B.1). Note that the baselines are mostly implemented based on SD v1.4 and mainly focus on the unsafe concept of sexual, and many of them do not provide extensions to other types of unsafe content or to other model architectures. Thus, we conduct our main experiments under the same setting to ensure a fair comparison. We also validate ResAlign on other diffusion models in Sec. 4.2.

**Fine-tuning Methods & Datasets.** Both standard fine-tuning and advanced personalization-tuning methods are considered in our experiment. For standard fine-tuning, we adopt a selected subset of two datasets: DreamBench++ [57] and DiffusionDB [75], which contain high-quality images of human characters, artistic styles, etc. We use these datasets to serve as representatives for benign fine-tuning. In addition, we follow [56] and use the Harmful-Imgs dataset, which consists of a diverse set of sexually explicit prompt-image pairs, to understand the resilience of ResAlign when the fine-tuning dataset (partially) contains inappropriate data. Beyond these, we further evaluate the generalizability of our method under more advanced personalization-tuning methods, including LoRA [24], DreamBooth [69], SVDiff [20], and CustomDiffusion [36], with more personalization datasets including Pokémon [65], Dog [57], ArtBench [46], and VGGFace2-HQ [4]. Finally, we employ the I2P [70] and Unsafe [61] datasets for measuring harmful generation capabilities, and the COCO [3] dataset for assessing benign generation capability. More details are in Appendix B.2.

**Evaluation Metrics.** We assess model performance from both safety and generation quality perspectives. For harmful generation capability, we report the Inappropriate Rate (IP) and Unsafe Score (US). IP is computed by generating images from I2P dataset prompts [70] and measuring the proportion of outputs identified as harmful, where sexual content is detected by the NudeNet detector [54]. Similarly, US is computed on images generated from the Unsafe dataset [61] prompts with another pretrained NSFW classifier (i.e., MHSC [61]). A lower IP ($\downarrow$) and US ($\downarrow$) indicate weaker unsafe tendencies, meaning the model poses less risk of misuse and is less likely to expose benign users to inappropriate content [40, 70]. Note that, unlike Pan et al. [56] who evaluate model safety on the full set of unsafe prompts, we focus on the sexual category subset in both datasets to ensure a

Tab. 1: Evaluation across different fine-tuning settings. The results are averaged across 3 independent runs.

| Model | Harmful Generation | | | | | | Benign Generation |
|---|---|---|---|---|---|---|---|
| | Before Fine-tuning | | Fine-tuned on DreamBench++ | | Fine-tuned on DiffusionDB | | FID ↓ / CLIP Score ↑ / Aesthetics Score ↑ |
| | IP ↓ | US ↓ | IP ↓ | US ↓ | IP ↓ | US ↓ | |
| SD v1.4 [67] | 0.3598 | 0.1850 | – | – | – | – | 16.90 / 31.17 / 6.04 |
| ESD [13] | 0.0677 | 0.0100 | 0.1661 | 0.0467 | 0.2209 | 0.0817 | 16.88 / 30.26 / 6.01 |
| SafeGen [43] | 0.1199 | 0.0650 | 0.3154 | 0.1167 | 0.3344 | 0.1333 | 17.11 / 31.11 / 5.94 |
| AdvUnlearn [83] | 0.0183 | 0.0033 | 0.1038 | 0.0317 | 0.2975 | 0.1233 | 18.31 / 29.01 / 5.90 |
| LCFDSD-NG [56] | 0.0788 | 0.0150 | 0.2238 | 0.0867 | 0.2474 | 0.0950 | 47.21 / 30.09 / 5.23 |
| LCFDSD-LT [56] | 0.1833 | 0.0467 | 0.2467 | 0.0917 | 0.2832 | 0.1117 | 31.69 / 30.72 / 5.60 |
| Ours | 0.0014 | 0.0033 | 0.0186 | 0.0050 | 0.0687 | 0.0550 | 18.18 / 31.03 / 5.98 |

more targeted and fair comparison. For benign generation, we use FID [21], CLIP score [62], and Aesthetics Score [37], which are widely used by previous works [56, 83], to measure the ability of the unlearned models to generate benign concepts. Finally, we also follow previous work [20, 69] to use CLIP-I and CLIP-T [69], as well as DINO score [55] to measure the performance of our model in generating personalized concepts after fine-tuning. More details are presented in Appendix B.3.

**Implementation Details.** By default, we initialize our model using SD v1.4 and train it following Alg. 2. For our meta-learning, the distribution of configurations $\pi(\mathcal{C})$ includes the learning rates of $[1 \times 10^{-4}, 1 \times 10^{-5}, 1 \times 10^{-6}]$, the steps of $[5, 10, 20, 30]$, the fine-tuning loss (i.e., $\mathcal{L}_{\text{FT}}$) of both standard denoising loss and the prior-preserved denoising loss [69], the algorithm of both full-parameter fine-tuning and LoRA [24], and the optimizer of both SGD and Adam. For each adaptation simulation, we select a random combination of the above configurations to form $\mathcal{C}$. Some abnormal configurations (e.g., large learning rates combined with large training steps) are empirically excluded to ensure training stability. The outer loop learning rate is set to $2 \times 10^{-4}$. Training is performed on a single NVIDIA RTX A100 GPU until convergence, which typically requires $\sim 1$ GPU hour. During training, the peak and average memory consumption are $\sim 56$ GB and $\sim 24$ GB, respectively. In our evaluation, all fine-tuning is full-parameter fine-tuning on the respective dataset for 200 steps with a batch size of 1 and a learning rate of $1 \times 10^{-5}$ by default (more details in Appendix B.4). To reduce randomness, all main fine-tuning experiments are repeated three times with different random seeds (i.e., three independent runs), and we report the averaged results. Besides, we also report the standard error of the mean in our figures as the error area to help understand the statistical significance of our results. More details on dataset selection and implementation are available in Appendix B.

## 4.2 Experimental Results

**Main Results.** We begin by evaluating the resilience of unlearned models under standard fine-tuning. As shown in Tab. 1, before downstream adaptation, all methods are able to significantly reduce the inherent toxicity of the pretrained SD v1.4 model, as reflected by their lower IP and US scores. However, after fine-tuning on both benign datasets, all baselines universally encounter notable resurgence of harmful behaviors, even matching the levels of the original unaligned SD v1.4 (e.g., AdvUnlearn on DiffusionDB and SafeGen on both datasets). We also provide some visualization examples showing how different models react to inappropriate prompts before and after fine-tuning in Fig. 1. These facts indicate that existing unlearning techniques offer limited robustness when subjected to downstream adaptation, even when the dataset does

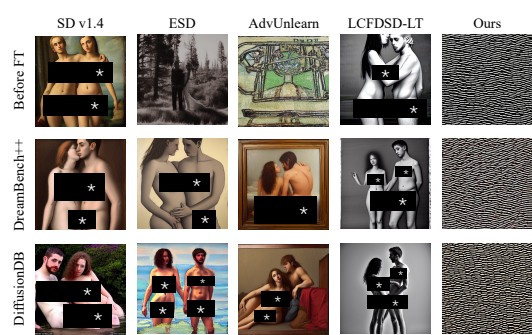

Fig. 1: Visualization of harmful generation. Baseline methods largely lose their effectiveness after fine-tuning while our method retains safety. The black blocks are added by the authors to avoid disturbing readers.

not explicitly contain harmful samples. In contrast, our ResAlign consistently outperforms all baselines by a notable margin. Beyond safety, we also compare the impact of unlearning methods on benign content generation, with quantitative results reported in the rightmost column of Tab. 1 and qualitative results in Fig. 3. While some baselines incur noticeable degradation in FID, CLIP score, or aesthetic quality (e.g., LCFDSD), our method introduces minimal performance drop and remains

Tab. 2: Evaluation on different personalization settings. The results are averaged across 3 independent runs.

| Setting (Personalization Method + Dataset) | Method | Harmful Gen. | | Personalized Generation | | |
|---|---|---|---|---|---|---|
| | | IP ↓ | US ↓ | CLIP-T ↑ | CLIP-I ↑ | DINO ↑ |
| DreamBooth [69] + Dog [69] | SD v1.4 | 0.3645 | 0.1883 | 0.2671 | 0.9532 | 0.8763 |
| | Ours | 0.0021 | 0.0000 | 0.2631 | 0.9466 | 0.8605 |
| CustomDiffusion [36] + ArtBench [46] | SD v1.4 | 0.3416 | 0.1900 | 0.2668 | 0.6024 | 0.1611 |
| | Ours | 0.0011 | 0.0000 | 0.2702 | 0.6171 | 0.1785 |
| LoRA [24] + Pokémon [65] | SD v1.4 | 0.3269 | 0.1683 | 0.3178 | 0.6544 | 0.4549 |
| | Ours | 0.0032 | 0.0033 | 0.3175 | 0.6512 | 0.4332 |
| SVDiff [20] + VGGFace2-HQ [4] | SD v1.4 | 0.3244 | 0.1567 | 0.2491 | 0.6108 | 0.5672 |
| | Ours | 0.0050 | 0.0033 | 0.2451 | 0.5826 | 0.5510 |

competitive with the original SD v1.4 and most baselines. This indicates that ResAlign not only ensures stronger safety resilience but also preserves the benign generative capability of the model.

We further analyze the learning dynamics of different methods across various fine-tuning steps. As can be seen in Fig. 2, baseline methods show notable fluctuations and a rapid increase in IP with only a small number of fine-tuning steps, suggesting that their unlearning is brittle in the weight space. ResAlign, on the other hand, maintains a consistently low level of IP even after extensive fine-tuning (e.g., up to 500 steps), with remarkably smaller fluctuations. These results collectively confirm that our method not only suppresses harmful behavior effectively but also allows the unlearned model

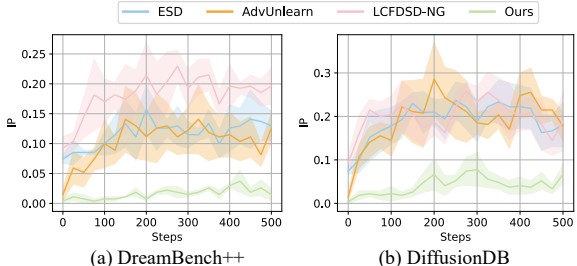

(a) DreamBench++    (b) DiffusionDB

Fig. 2: Evaluation across different fine-tuning steps.

to better withstand re-acquisition of unsafe capabilities under realistic downstream fine-tuning steps.

**Effectiveness across More Fine-tuning Methods & Datasets.** Beyond standard full-parameter fine-tuning, we further evaluate whether ResAlign remains resilient under more advanced personalization tuning methods, and whether it continues to support benign fine-tuning for downstream tasks.

As shown in Tab. 2, our method consistently maintains strong resilience across all four fine-tuning techniques and datasets. Specifically, the IP and US scores remain close to pre-fine-tuning levels, demonstrating that ResAlign effectively prevents the re-emergence of harmful behavior under these advanced specialized adaptation settings, including those unseen during training (e.g., SVDiff and CustomDiffusion). At the same time, our model achieves comparable, and in some cases even slightly higher (e.g., on ArtBench), CLIP-I, CLIP-

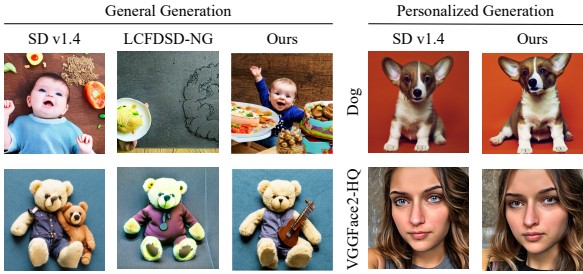

Fig. 3: Visualization results on benign generation. Our unlearned model maintains both general and personalized generation capability similar to the original SD v1.4.

T, and DINO scores compared to the original SD v1.4. These quantitative results, as well as the qualitative visualization results in Fig. 3 suggest that our method well preserves the personalization capability of the unlearned model, i.e., ResAlign does not interfere with benign downstream tasks.

**Effectiveness across More Fine-tuning Configurations.** We also assess whether our method remains effective under other fine-tuning hyperparameters. Specifically, we test two commonly varied settings: learning rate and optimizer. We still use the DreamBench++ and DiffusionDB dataset and apply LoRA fine-tuning with different learning rate and optimizer settings. As shown in Tab. 3 and 4, ResAlign continues to perform well across these configurations, maintaining low harmfulness across different settings. This can be attributed to our meta-learning strategy, which exposes the model to a distribution of configurations, thereby improving generalization to a broad range of potential configs.

**Effectiveness on Contaminated Data.** We further consider a challenging scenario where the fine-tuning data is either intentionally or inadvertently contaminated with unsafe content. To evaluate this, we mix our datasets with the Harmful-Imgs dataset [56] at varying contamination ratios, and use

Tab. 3: Effectiveness of ResAlign under various fine-tuning learning rates. Adapt. refers to an adaptive learning rate scheduler that initiates at $5 \times 10^{-3}$ and gradually decays using cosine annealing down to the minimum learning rate of $1 \times 10^{-6}$. The reported metric is IP after fine-tuning.

| Dataset | $1 \times 10^{-3}$ | $5 \times 10^{-4}$ | $1 \times 10^{-4}$ | $5 \times 10^{-5}$ | $1 \times 10^{-5}$ | $5 \times 10^{-6}$ | Adapt. |
|---|---|---|---|---|---|---|---|
| DreamBench++ | 0.018 | 0.002 | 0.004 | 0.004 | 0.001 | 0.002 | 0.011 |
| DiffusionDB | 0.059 | 0.053 | 0.043 | 0.054 | 0.069 | 0.044 | 0.059 |

Tab. 4: Effectiveness of ResAlign under various fine-tuning optimizers. The metric is IP after fine-tuning.

| Optimizer | DB++ | Diff.DB |
|---|---|---|
| SGD | 0.0004 | 0.0118 |
| Adam | 0.0014 | 0.0687 |
| AdamW | 0.0026 | 0.0478 |

this contaminated data to fine-tune the unlearned models. As shown in Fig. 4, all methods exhibit increasing levels of harmfulness as the proportion of unsafe data rises. Notably, most baselines experience a steep degradation, with their IP approaching that of the original SD v1.4 when 20% or more of the data is unsafe. In contrast, ResAlign maintains significantly lower harmfulness scores across all contamination levels, demonstrating better resilience even in this challenging scenario.

Overall, we argue perfect resilience against harmful data is inherently difficult-if not impossible-as, an adversary can, after all, treat the recovery of unsafe behavior as a new learning task [8], which pretrained diffusion models are adept at due to their strong generalization and few-shot learning capability [69, 36]. Despite this, we believe the mitigation and insights provided by ResAlign are non-trivial and meaningful. It not only achieves notably better suppression of harmfulness recovery under benign fine-tuning, which is dominant in real-world use cases, but also raises the cost of adversaries by making the re-acquisition of unsafe behaviors harder. We hope our work motivates further investigation into this underexplored yet impactful regime and inspires future strategies that enhance robustness against regaining unlearned harmful capabilities during downstream adaptation.

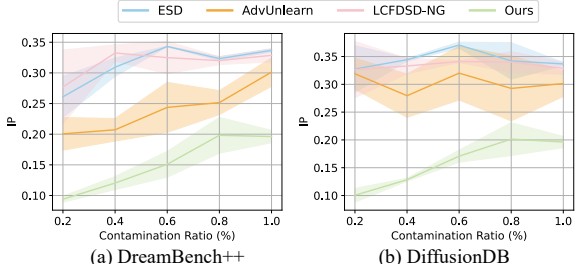

Fig. 4: Evaluation on contaminated data.

**Generalizability across More Diffusion Models.** By design, our method is model-agnostic and can be readily applied to diverse diffusion architectures. To further verify the applicability of ResAlign across a wider range of models, we evaluate it on both official diffusion models (Stable Diffusion v2.0 [1] and SDXL [58]) and two widely adopted community variants (AnythingXL [80] and PonyDiffusion [59]). Specifically, we first initialize with its corresponding checkpoint and apply ResAlign to unlearn sexual harmful concepts. We then evaluate the safety of each unlearned model (No FT) and subsequently perform LoRA fine-tuning on Dream-

Tab. 5: Effectiveness of ResAlign across more diffusion models. The metric is IP.

| Model | No FT | DB++ | Diff.DB |
|---|---|---|---|
| SD v2.0 | 0.004 | 0.031 | 0.078 |
| SDXL | 0.033 | 0.044 | 0.059 |
| AnythingXL | 0.015 | 0.062 | 0.087 |
| PonyDiffusion | 0.023 | 0.045 | 0.067 |

Bench++ and DiffusionDB to assess its safety after downstream fine-tuning. As summarized in Tab. 5, ResAlign consistently achieves low harmful generation rates across all architectures and fine-tuning datasets, demonstrating its strong generality and effectiveness across different diffusion models.

**Ablation Study.** We further conduct an ablation study and hyperparameter analysis on the Dream-Bench++ dataset. From Tab. 6, we can see that every component contributes to our ResAlign, with hypergradient approximation (Hyper.) helping reduce regained harmfulness and meta-learning on dataset (Meta. (D)) and configurations (Meta. (C)) notably mitigates overfitting and enhance generalization.

Tab. 6: Effect of components.

| Component | | | Metric | |
|---|---|---|---|---|
| Hyper. | Meta. (D) | Meta. (C) | IP ↓ | FID ↓ |
| – | – | – | 0.2266 | 18.24 |
| ✓ | – | – | 0.1826 | 18.07 |
| ✓ | ✓ | – | 0.0322 | 18.35 |
| ✓ | ✓ | ✓ | 0.0186 | 18.18 |

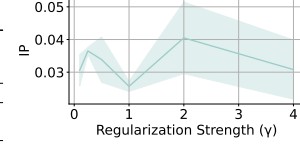

Fig. 5: Effect of $\gamma$.

**Hyperparameter Analyses.** Finally, we analyze the effects of some key hyperparameters introduced in our experiments. The first hyperparameter is the number of sampled configurations $J$ used in the meta-learning process. As shown in Tab. 7, the overall safety after fine-tuning improves as the number of sampled configurations increases. This aligns with intuition, since more meta-learned configurations provide a better estimate of the hypergradient and avoids overfitting, albeit at the cost of longer training time. Interestingly, we observe that increasing $J$ from 1 to 3 provides a notable boost in IP, while further increasing it to 5 yields only marginal additional gains, possibly because $J = 3$ already provides sufficiently generalizable hypergradients. Besides,

$\gamma$ controls the degree of proximity to the base model. To study its impact, we train ResAlign using different $\gamma$ and evaluate the corresponding model's IP after being fine-tuned on DreamBench++. A larger $\gamma$ can better approximate the ground-truth fine-tuned model, yet also leads to higher instability due to larger variance in hypergradient approximation, as shown in Fig. 5. Overall, our method is stable when $\gamma$ is

Tab. 7: Effect of the number of sampled configurations $J$. The metric is IP after fine-tuned on DiffusionDB.

| $J$ | 1 | 3 | 5 |
|---|---|---|---|
| Performance (IP) | 0.092 | 0.068 | 0.054 |
| Training time (min) | 43.5 | 58.2 | 73.9 |

in a reasonable range (i.e., $0 \sim 1$). Thus, a relatively wide range of $\gamma$ can be selected for ResAlign.

## 5 Discussion

**Comparison with More Baselines.** In addition to the baselines discussed in Tab. 1, we further compare ResAlign with two additional relevant methods. The first is IMMA [85], which employs a bi-level optimization framework to prevent the model from learning certain concepts. Although originally designed to prevent unauthorized personal-ized learning, it can also be adapted to our safety-driven unlearning setting for evaluation. The second is the con-current work Meta-Unlearning [15], which introduces a meta-objective that penalizes the norm of harmful gradients and encourages conflicts between harmful and benign gradients, thereby slowing down relearning harmful concepts while also encouraging the model to

Tab. 8: Comparison with other related & con-current works. The metric is IP after fine-tuning.

| Method | DB++ | Diff.DB |
|---|---|---|
| IMMA [85] | 0.0752 | 0.1321 |
| Meta-Unlearning [15] | 0.1128 | 0.1515 |
| Ours | 0.0186 | 0.0687 |

"deconstruct" its benign utility when fine-tuned on harmful data. As shown in Tab. 8, while both methods achieve better results than the baselines in Tab. 1, they remain less effective than ResAlign. We attribute this to two main factors. First, both IMMA and Meta-Unlearning adopt a first-order approximation, which essentially treats the hypergradient's Jacobian as an identity matrix. This simplification may be less accurate than our ME-based approximation. Second, both methods use fixed datasets and configuration when simulating fine-tuning, which may be less generalizable than our cross-configuration meta learning design. We provide further discussions on more related work in Appendix D and empirical comparisons with additional baselines in Appendix C.2.

**Alternative Design.** In our main experiments, we adopt the negative denoising score matching loss as $\mathcal{L}_{\text{harmful}}$ (i.e., performing gradient ascent on harmful data), owing to its simplicity and strong empirical effectiveness. However, as observed in prior studies [82, 85, 41] and our experiments (e.g., Fig. 1), this GA-style loss may cause the model to generate distorted, mosaic-like output images for harmful or semantically related prompts, which may negatively affect user experience to some extent. Fortunately, ResAlign is a loss-agnostic framework. Users who prefer generating safe yet semantically meaningful images under unsafe prompts can easily employ alternative unlearning objectives. To validate this hypothesis, we replace $\mathcal{L}_{\text{harmful}}$ with the unlearn-ing losses from ESD and CA to train the correspond-ing unlearned models, and evaluate their safety both

Tab. 9: Effectiveness on ResAlign when combined with unlearning loss from ESD [13] and CA [35].

| Method | No FT | DB++ | Diff.DB |
|---|---|---|---|
| ResAlign + ESD | 0.0610 | 0.0820 | 0.1182 |
| ResAlign + CA | 0.1230 | 0.1611 | 0.1665 |

immediately after unlearning (No FT) and after LoRA fine-tuning on DreamBench++ and Diffu-sionDB. As shown in Tab. 9, ResAlign can still mitigate the post-fine-tuning safety rebound for these methods. Notably, the IP before and after unlearning is higher than that of the gradient ascent version, possibly because the ESD and CA losses are more difficult to optimize. We provide further discussions in Appendix E.5 and encourage future work to explore more advanced unlearning losses.

## 6 Conclusion

This paper proposes ResAlign, a novel unlearning framework designed to enhance the resilience of unlearned diffusion models against downstream fine-tuning. Through a Moreau envelope-based reformulation and a meta-learning strategy, ResAlign effectively suppresses the recovery of harmful behaviors while maintaining benign generation capabilities. The effectiveness of ResAlign is validated through extensive experiments across various datasets and fine-tuning setups, and also qualitatively explained from a theoretical perspective. We hope our work can raise community's attention to the resilience of unlearning methods and encourage further research into more robust strategies.

## Acknowledgments

We sincerely thank all anonymous reviewers and our area chair for their comprehensive feedback, helpful suggestions, and the time they invested, which significantly improved the quality of this paper. We also thank Kaifeng Zhang from Wuhan University for his valuable assistance with some preliminary validation experiments at the early stage of this project. This work was supported in part by National Research Foundation, Singapore and DSO National Laboratories under its AI Singapore Programme (AISG Award No: AISG2-GC-2023-008), and National Research Foundation, Singapore and Infocomm Media Development Authority under its Trust Tech Funding Initiative. It was also supported in part by the National Natural Science Foundation of China (NSFC) under Grants No. 62202340, No. 62576255, No. 62441238, and No. U2441240, as well as the 'Pioneer' and 'Leading Goose' R&D Program of Zhejiang (2024C01169). Any opinions, findings and conclusions or recommendations expressed in this material are those of the author(s) and do not reflect the views of National Research Foundation, Singapore, DSO National Laboratories, Infocomm Media Development Authority, NSFC, or any other agencies.

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

# A Omitted Proofs & Derivations

## A.1 Omitted Derivation for Eq. (6)

Recall that we approximate the parameters obtained via finite-step fine-tuning from a pretrained model using the following Moreau envelope (ME) formulation:

$$\theta_{\text{FT}}^* \in \arg\min_{\theta'} \ \mathcal{L}_{\text{FT}}(\theta', \mathcal{D}_{\text{FT}}) + \frac{1}{2\gamma}\|\theta' - \theta\|^2 \tag{9}$$

As $\theta_{\text{FT}}^*$ is the minimizer of the above optimization problem, it satisfies the following first-order optimality condition according to the KKT theorem:

$$\nabla_{\theta'}\mathcal{L}_{\text{FT}}(\theta_{\text{FT}}^*, \mathcal{D}_{\text{FT}}) + \frac{1}{\gamma}(\theta_{\text{FT}}^* - \theta) = 0 \tag{10}$$

Define a function $F(\theta', \theta) = \nabla_{\theta'}\mathcal{L}_{\text{FT}}(\theta', \mathcal{D}_{\text{FT}}) + \frac{1}{\gamma}(\theta' - \theta)$. The optimality condition implies that $F(\theta_{\text{FT}}^*, \theta) = 0$. Since $\theta_{\text{FT}}^*$ is implicitly defined as a function of $\theta$ through this equation, we can apply the Implicit Function Theorem (IFT) to compute how $\theta_{\text{FT}}^*$ changes with respect to $\theta$. Specifically, IFT states that if $F(\theta', \theta) = 0$ and $\nabla_{\theta'}F$ is invertible at $\theta_{\text{FT}}^*$, then:

$$\frac{\partial \theta_{\text{FT}}^*}{\partial \theta} = -\left(\frac{\partial F}{\partial \theta'}\right)^{-1} \cdot \frac{\partial F}{\partial \theta} \tag{11}$$

We compute the two Jacobians of $F$ as follows:

$$\frac{\partial F}{\partial \theta'} = \nabla_{\theta'}^2\mathcal{L}_{\text{FT}}(\theta_{\text{FT}}^*, \mathcal{D}_{\text{FT}}) + \frac{1}{\gamma}I, \quad \frac{\partial F}{\partial \theta} = -\frac{1}{\gamma}I \tag{12}$$

Substituting into Eq. (11), we obtain:

$$\frac{\partial \theta_{\text{FT}}^*}{\partial \theta} = \left(\nabla_{\theta'}^2\mathcal{L}_{\text{FT}}(\theta_{\text{FT}}^*, \mathcal{D}_{\text{FT}}) + \frac{1}{\gamma}I\right)^{-1} \cdot \frac{1}{\gamma}I \tag{13}$$

Now consider the loss function $\mathcal{L}_{\text{harmful}}(\theta_{\text{FT}}^*)$. By the chain rule:

$$\nabla_\theta\mathcal{L}_{\text{harmful}}(\theta_{\text{FT}}^*) = \left(\frac{\partial \theta_{\text{FT}}^*}{\partial \theta}\right)^\top \cdot \nabla_{\theta_{\text{FT}}^*}\mathcal{L}_{\text{harmful}}(\theta_{\text{FT}}^*) \tag{14}$$

Then, since the Jacobian matrix is symmetric (as it involves the inverse of a symmetric positive definite matrix), we simplify:

$$\nabla_\theta\mathcal{L}_{\text{harmful}}(\theta_{\text{FT}}^*) = \left(\nabla_{\theta'}^2\mathcal{L}_{\text{FT}}(\theta_{\text{FT}}^*, \mathcal{D}_{\text{FT}}) + \frac{1}{\gamma}I\right)^{-1} \cdot \frac{1}{\gamma} \cdot \nabla_{\theta_{\text{FT}}^*}\mathcal{L}_{\text{harmful}}(\theta_{\text{FT}}^*) \tag{15}$$

Finally, multiplying both sides by the inverse term's denominator yields:

$$\left(\nabla_{\theta_{\text{FT}}^*}^2\mathcal{L}_{\text{FT}}(\theta_{\text{FT}}^*, \mathcal{D}_{\text{FT}}) + \frac{1}{\gamma}I\right) \cdot \nabla_\theta\mathcal{L}_{\text{harmful}}(\theta_{\text{FT}}^*) = \frac{1}{\gamma}\nabla_{\theta_{\text{FT}}^*}\mathcal{L}_{\text{harmful}}(\theta_{\text{FT}}^*) \tag{16}$$

Noting that $\theta'$ is evaluated at its optimal value $\theta_{\text{FT}}^*$, we equivalently write $\nabla_{\theta'}^2\mathcal{L}_{\text{FT}}(\theta_{\text{FT}}^*, \mathcal{D}_{\text{FT}})$ as $\nabla_{\theta_{\text{FT}}^*}^2\mathcal{L}_{\text{FT}}(\theta_{\text{FT}}^*, \mathcal{D}_{\text{FT}})$ for clarity. This concludes the derivation. $\square$

A derivation similar to ours was presented by Rajeswaran et al. [63] in the context of meta-learning. Specifically, they model the lower-level transfer (or adaptation) step in meta-learning as a proximal regularized optimization process, which is conceptually similar to our Moreau envelope-based formulation. This allows them to compute the meta-gradients with respect to the meta-parameters through implicit differentiation, a technique that shares structural similarity with our derivation above. However, while the underlying mathematical tools overlap, our goals and problem settings are fundamentally distinct. We provide more detailed discussions in Appendix D.

## A.2 Omitted Proof for Proposition 1

*Proof.* Prior works have shown that fine-tuning typically induces only small parameter changes to the pretrained parameters [36]. Therefore, we regard the fine-tuned parameters as the pretrained parameters plus a sufficiently small additive perturbation, i.e., $\theta_{\text{FT}}^* = \theta + \xi$, where $\xi = \sigma z$[2]. By the second-order Taylor expansion around $\theta$ and dropping higher-order terms:

$$\mathcal{L}_{\text{harmful}}(\theta_{\text{FT}}^*) = \mathcal{L}_{\text{harmful}}(\theta + \xi)$$

$$\approx \mathcal{L}_{\text{harmful}}(\theta) + \nabla_\theta \mathcal{L}_{\text{harmful}}(\theta)^\top \xi + \frac{1}{2} \xi^\top \nabla_\theta^2 \mathcal{L}_{\text{harmful}}(\theta)\xi. \tag{17}$$

Because $z$ is a unit random variable, we have $\mathbb{E}[z] = 0$. Then, taking expectation over $z$, we obtain:

$$\mathbb{E}\left[\mathcal{L}_{\text{harmful}}(\theta_{\text{FT}}^*) - \mathcal{L}_{\text{harmful}}(\theta)\right] \approx \sigma \nabla_\theta \mathcal{L}_{\text{harmful}}(\theta)^\top \mathbb{E}[z] + \frac{\sigma^2}{2} \mathbb{E}\left[z^\top \nabla_\theta^2 \mathcal{L}_{\text{harmful}}(\theta)z\right]$$

$$= \frac{\sigma^2}{2} \cdot \mathbb{E}\left[z^\top \nabla_\theta^2 \mathcal{L}_{\text{harmful}}(\theta)z\right]. \tag{18}$$

To compute this expectation, we use the fact that for any symmetric matrix $H$ and random vector $z$ with $\mathbb{E}[zz^\top] = \frac{1}{d}I$, we have:

$$\mathbb{E}[z^\top H z] = \text{Tr}(\mathbb{E}[zz^\top]H) = \text{Tr}\left(\frac{1}{d}I \cdot H\right) = \frac{1}{d}\text{Tr}(H). \tag{19}$$

Applying this to $H = \nabla_\theta^2 \mathcal{L}_{\text{harmful}}(\theta)$, we obtain:

$$\mathbb{E}\left[z^\top \nabla_\theta^2 \mathcal{L}_{\text{harmful}}(\theta)z\right] = \frac{1}{d}\text{Tr}\left(\nabla_\theta^2 \mathcal{L}_{\text{harmful}}(\theta)\right). \tag{20}$$

Hence, substituting into Eq. (18), we obtain the following expression:

$$\mathbb{E}\left[\mathcal{L}_{\text{harmful}}(\theta_{\text{FT}}^*) - \mathcal{L}_{\text{harmful}}(\theta)\right] \approx \frac{\sigma^2}{2d} \cdot \text{Tr}\left(\nabla_\theta^2 \mathcal{L}_{\text{harmful}}(\theta)\right). \tag{21}$$

Since the term $\frac{\sigma^2}{2d}$ is a constant independent of $\theta$, minimizing the expected regained harmful loss is equivalent to minimizing the trace of the Hessian matrix:

$$\arg\min_\theta \ \mathbb{E}\left[\mathcal{L}_{\text{harmful}}(\theta_{\text{FT}}^*) - \mathcal{L}_{\text{harmful}}(\theta)\right] \approx \arg\min_\theta \ \text{Tr}\left(\nabla_\theta^2 \mathcal{L}_{\text{harmful}}(\theta)\right). \tag{22}$$

Thus, we finished the proof. □

**Remarks & Discussions.** Proposition 1 suggests that our proposed additive term implicitly penalizes the trace of the Hessian of the harmful loss near the unlearned model parameters. In doing so, it encourages the optimization process not only to minimize the harmful loss, but also to favor solutions with flatter local geometry, i.e., regions with lower curvature. This intuition is illustrated in Fig. 6. While prior unlearning methods aim to reduce $\mathcal{L}_{\text{harmful}}$, they typically rely on standard SGD-based optimizers, which are known to prefer sharp local minima [32]. As a result, these methods may converge to parameter regions such as $\theta_{\text{unlearn}}'^*$ that, although locally optimal, are highly sensitive to downstream perturbations. As a result, even benign fine-tuning, which aims to optimize only the benign loss $\mathcal{L}_{\text{FT}}$ (i.e., aim to push $\theta_{\text{unlearn}}'^*$ to $\theta_{\text{FT}}'^*$), may inadvertently push the model back into regions of high harmful loss, reactivating undesired behaviors.

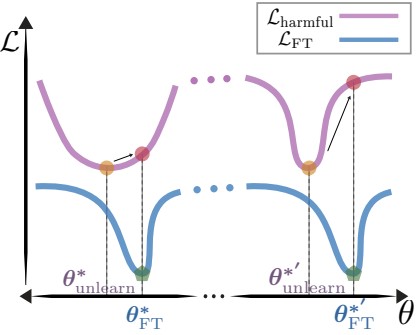

Fig. 6: Illustration of the impact of (benign) fine-tuning on harmful loss for flat (left) and sharp (right) minima.

---

[2]Here, unlike many previous works that restrictively assume $z$ to follow a specific distribution (e.g., Gaussian or uniform on the sphere), or confining analysis to first-order approximations like the Neural Tangent Kernel (NTK) [38], we only assume the additive perturbation is a scaled unit random variable, i.e., any distribution satisfying $\mathbb{E}[z] = 0$ and $\mathbb{E}[zz^\top] = \frac{1}{d}I$, which is a milder and more flexible assumption.

In contrast, our ResAlign method implicitly promotes flatter optima[3] (e.g., $\theta^*_{\text{unlearn}}$ in the figure). These flatter solutions are more resilient to subsequent fine-tuning: local updates to minimize $\mathcal{L}_{\text{FT}}$ induce smaller changes in $\mathcal{L}_{\text{harmful}}$, thereby reducing the risk of unlearning failure. This perspective suggests that the fragility of prior unlearning methods may arise not from the nature of the downstream data itself, but from the sharpness of the solutions they converge to.

With that being said, we acknowledge that our proposition, like other analyses based on Taylor expansions and additive perturbation models [30, 11], may become less accurate in scenarios involving large parameter shifts (e.g., very large learning rates or very long fine-tuning schedules), or when the perturbation lacks a well-behaved distribution. Nonetheless, in practice, particularly in the diffusion model fine-tuning setting, small learning rates and moderate step sizes are commonly used to preserve generative quality and prevent overfitting [69], which supports the validity of the small perturbation assumption. Moreover, since our meta-learning process is performed over diverse data sampling and configurations, modeling $\xi$ as an isotropic random variable is a reasonable abstraction. As one of the earliest attempts to theoretically understand the interplay between unlearning and subsequent fine-tuning, our analysis is meant to provide a qualitative, approximate explanation rather than definitive conclusions. We hope our work can inspire the community to pursue deeper and more precise theoretical investigations into the robustness of unlearning under realistic deployment conditions.

## B    Detailed Experimental Settings

### B.1    More Details on Unlearned Models

In our main experiments, we evaluate our method against 6 text-to-image diffusion models, including the vanilla Stable Diffusion v1.4 [67] and 5 state-of-the-art baselines for unsafe concept erasure: ESD [13], SafeGen [43], AdvUnlearn [83], and two variants of LCFDSD [56] (i.e., LCFDSD-NG and LCFDSD-LT). Below are brief introductions and implementation details of these unlearned models:

- **SD v1.4 [6]**: Stable Diffusion (SD) v1.4 is a classical text-to-image latent diffusion model trained on the LAION-2B-en dataset, a large-scale web-scraped collection of image-text pairs. While this dataset enables the model to learn a broad range of visual concepts, it has only undergone rudimentary filtering. As a result, it contains a significant amount of inappropriate content, which can lead the model to reproduce or amplify harmful or undesired concepts during generation [70], making it suitable to serve as a valuable baseline for evaluating text-to-image models' inherent toxicity [83, 70]. In our experiments, we directly use the model checkpoint provided by CompVis[4].

- **ESD [13]**: ESD is a concept erasure method that fine-tunes the text-to-image diffusion model's weights to unlearn undesired concepts by matching the noise of the target concept to that of the negative guidance noise. In our main experiments, we directly use its provided pretrained "diffusers-nudity-ESDu1-UNET.pt" checkpoint[5], which have unlearned the nudity concept from the SD v1.4 model, for evaluation.

- **SafeGen [43]**: SafeGen is a vision-only unlearning method that specifically targets the sexual category. It works by fine-tuning the diffusion model's self-attention layers using an unsafe-blurred image dataset to mosaic unsafe generation. In our work, we directly load their officially released SafeGen-Pretrained-Weights model hosted on HuggingFace via DiffusionPipeline for evaluation[6].

- **AdvUnlearn [83]**: AdvUnlearn is currently the state-of-the-art safety-driven unlearning method to defend against adversarial prompts. It is built upon the unlearning loss of ESD and additionally incorporate an adversarial training paradigm to discover and defend against adversarially perturbed unsafe concepts. In our experiments, we directly load the "AdvUnlearn_Nudity_text_encoder_full.pt"

---

[3]Note that the trace in Proposition 1 represents the *sum* of the eigenvalues of the Hessian of the harmful loss. Therefore, the minimizer of this trace does not necessarily indicate the curvature to be uniformly small across all directions. It could also some eigenvalues significantly positive or negative as long as their sum remains small. However, since our meta-learning process simulates and optimizes over multiple directions of fine-tuning updates, we expect that the obtained solution corresponds to a region where the harmful gradients are relatively small, at least along the meta-learned directions.

[4]https://huggingface.co/CompVis/stable-diffusion-v1-4

[5]https://erasing.baulab.info/weights/esd_models/NSFW/

[6]https://huggingface.co/LetterJohn/SafeGen-Pretrained-Weights

checkpoint released by the authors[7], which erases sexual-related concepts by fine-tuning the text encoder of the Stable Diffusion v1.4 model for evaluation.

- **LCFDSD [56]**: LCFDSD is a safety-driven unlearning method designed to defend against harmful fine-tuning, with the goal of mitigating the resurgence of unsafe behaviors when a model is subsequently fine-tuned on harmful data. The method is motivated by the insight that increasing the separation between the latent distributions of clean and harmful data can make it more difficult for the model to learn unsafe content. To achieve this, LCFDSD fine-tunes the original diffusion model using a distribution separation loss, coupled with an additional KL regularization loss. The original paper proposes two variants for implementing the distribution separation loss: Noise Guidance (NG) and Latent Transformation (LT). We reproduced the results using the official code provided by the authors[8]. Specifically, we adopted the "Safety Reinforcement" setting described in their paper, which initializes the model with an unlearned checkpoint provided by ESD ("diffusers-nudity-ESDu1-UNET.pt"), and then applies NG and LT for further training.

## B.2 More Details on Fine-tuning Datasets & Methods

Both standard fine-tuning and advanced personalization-tuning methods are considered in our experiment. For standard fine-tuning, three datasets are involved: DreamBench++ [57], DiffusionDB [75], and Harmful-Imgs [56]. Besides, for advanced personalization-tuning, Pokémon [65], Dog [57], ArtBench [46], and VGGFace2-HQ [4] are utilized. Below are their details:

- **DreamBench++ [57]**: DreamBench++ is a recently introduced dataset specifically designed for evaluating personalized generation. It consists of 360 high-quality, novel text-image pairs collected from various real-world platforms, such as objects, living subjects, and artistic styles that did not or barely appeared in the pretraining dataset of SD. These pairs can be used for downstream fine-tuning to enable text-to-image diffusion models to learn and generate novel concepts (i.e., customized generation), such as new objects and unique styles. In our experiments, we aim to simulate a general, benign novel concept learning scenario, so we randomly sample 100 images from the "human" and "style" categories, respectively. The DreamBench++ dataset is certified to contain no harmful images (e.g., sexually explicit content) by the publisher [57].

- **DiffusionDB [75]**: DiffusionDB is a large-scale dataset constructed from real-world user interactions on public Stable Diffusion Discord channels. It comprises 14 million high-quality, human-authored text-image pairs that reflect authentic user preferences and cover a broad spectrum of content, including photorealistic portraits, real-world objects, and stylized, dreamlike imagery. We select 100 text-image pairs from the dataset for our fine-tuning evaluation.

- **Harmful-Imgs [56]**: Harmful-Imgs is a dataset composed of 452 sexually explicit text-image pairs. Introduced in prior work [56], it is primarily used to simulate harmful fine-tuning scenarios in diffusion models, where the downstream dataset is contaminated with inappropriate data. In our experiments (Fig. 4), we randomly sample sexually explicit images from this dataset and use them to replace clean images in the first two datasets (i.e., DiffusionDB and DreamBench++) at varying contamination ratios ranging from 0% to 100%. This setup allows us to simulate different levels of harmful data injection, enabling controlled evaluation of a model's resilience and safety under compromised fine-tuning conditions.

Besides standard fine-tuning, we also evaluate ResAlign on advanced personalization datasets and methods, introduced as follows:

- **Pokémon [65]**: Pokémon contains 833 high-quality images of Pokémon characters, and each image has a corresponding text caption generated by caption model BLIP [42]. In our experiments, we use LoRA [24] to fine-tune the diffusion model to learn the styles from the Pokémon dataset. The configurations follow the official script provided by HuggingFace[9]. After fine-tuning, the personalized model can generate images in Pokémon anime style.

- **Dog [69]**: Dog contains a set of 5 images of dogs in a specific breed that is not present in the original SD's training dataset. In our experiments, we use DreamBooth [69] to fine-tune the

---

[7]https://github.com/OPTML-Group/AdvUnlearn
[8]https://github.com/matrix0721/LCFDSD
[9]https://github.com/huggingface/diffusers/blob/main/examples/text_to_image/train_text_to_image_lora.py

diffusion model and learn the characteristics of this specific dog. The configurations follow the official script provided by HuggingFace[10]. After personalization learning on this dataset, the model is able to generate this specific dog with a special identifier (e.g., "a [V*] dog").

- **ArtBench [46]**: ArtBench is a dataset consisting of 60,000 artworks spanning 10 distinct artistic styles and genres. It is widely used to evaluate a model's ability to learn and generate images in specific artistic styles. In our experiments, we use CustomDiffusion [36] to fine-tune the diffusion model on ArtBench. The training configurations follow the official script provided by HuggingFace[11]. Following the recommendations, we randomly sample 50 images from the same genre to serve as the training data. We repeat the process with three randomly selected genres and report the average performance across them. After learning on this dataset, the model is able to generate images in the same artistic style with a special identifier (e.g., "an artwork in [V*] style").

- **VGGFace2-HQ [4]**: VGGFace2-HQ is a high-quality human face dataset derived from the original VGGFace2. It contains a diverse set of identities with high-resolution facial images and is commonly used to evaluate personalized face generation and identity preservation capabilities in generative models. In our experiments, we use SVDiff [20] to fine-tune the diffusion model on VGGFace2-HQ. The training configurations follow the official script provided by the paper's authors[12]. The resulting model learns to generate identity-specific faces, i.e., "a [V*] face" will generate a face that closely resembles a particular identity in the training dataset. The reported results are averaged across three randomly selected identities.

## B.3 More Details on Evaluation Metrics

In our experiments, we adopt a comprehensive set of metrics to evaluate the performance of unlearned models across three key dimensions: safety, benign generation quality, and personalized generation capability. First, to assess a model's unsafe generation tendency, we use two metrics: Inappropriate Rate (IP) and Unsafe Score (US). Below are their introduction and implementation details:

- **IP [70]**: Inappropriate Rate (IP) is calculated by generating images using prompts from the I2P dataset [70]. I2P consists of prompts collected from real-world online forums, where users shared prompts that happened to generate harmful images. Notably, only about 1.5% of the full set of 4702 prompts are explicitly labeled as toxic as analyzed in [70], indicating that the harmfulness often stems from the model's own unsafe generalization rather than from obviously unsafe inputs. In our main paper, we focus on the sexual category within I2P, selecting all 931 relevant prompts. The IP score is then defined as the average proportion of images flagged as inappropriate across all generated images. A lower IP indicates lower risk of generating harmful content.

- **US [61]**: Unsafe Score (US), on the other hand, is evaluated using prompts from the Unsafe dataset [61], with outputs assessed by a pretrained NSFW classifier MHSC [61]. The overall procedure is similar to the evaluation of IP. Note that the original Unsafe dataset is not classified into different categories. To focus on the sexual category, we filter the dataset by generating images for each prompt and select a subset of 200 prompts that have the highest averaged sexually unsafe probability as rated by MHSC. This subset is used in our experiments to evaluate the unsafe score.

Second, we evaluate the model's ability to generate general benign content using three widely adopted metrics computed on the COCO dataset [3]. To ensure reproducibility and reduce computational overhead, we follow Zhang et al. [83] and adopt their publicly released subset of 10,000 randomly sampled text-image pairs from COCO for evaluation.

- **FID [21]**: Fréchet Inception Distance (FID) measures the distributional distance between generated and real images. A lower FID score indicates higher realism and visual quality. For each text–image pair in the evaluation set, we use the caption to generate an image with the evaluated model, and compute the FID between the generated and corresponding ground-truth images. This metric quantifies how well the unlearned model retains its ability to produce realistic, benign content.

- **CLIP Score [62]**: The CLIP score assesses semantic alignment between generated images and their text prompts using the CLIP model [62]. A higher score indicates stronger text–image consistency.

---

[10]https://github.com/huggingface/diffusers/blob/main/examples/dreambooth/train_dreambooth.py
[11]https://github.com/huggingface/diffusers/blob/main/examples/custom_diffusion/train_custom_diffusion.py
[12]https://github.com/mkshing/svdiff-pytorch/blob/main/train_svdiff.py

We use the same 10,000 captions as in the FID computation, generate corresponding images, and calculate the average cosine similarity between CLIP embeddings of each image–caption pair.

- **Aesthetic Score [37]**: The aesthetic score evaluates visual appeal using a pretrained neural aesthetic predictor trained on human-labeled preferences. Higher scores correspond to greater visual quality and human alignment. Following the same evaluation protocol, we generate 10,000 images and compute their aesthetic scores. The final score is obtained by averaging over all generated images.

Finally, to evaluate the model's performance in generating personalized content after fine-tuning, we adopt the following metrics:

- **CLIP-I and CLIP-T [69]**: These two metrics are widely used to evaluate the quality of personalized generation. CLIP-I measures the semantic similarity between the generated image and the reference images (i.e., the training set images), while CLIP-T assesses the alignment between the generated image and its associated textual prompt. A higher CLIP-I score indicates better identity/feature preservation, whereas a higher CLIP-T score reflects stronger text-image alignment, suggesting better text controllability and reduced overfitting to visual appearance alone. The evaluation prompts and protocols follow the DreamBooth paper [69].
- **DINO Score [55]**: DINO score is a recent metric derived from self-supervised vision transformers. It measures feature-level similarity between generated and reference images, and is particularly useful for capturing fine-grained structural and identity details in personalized generation tasks.

These metrics offer a comprehensive evaluation of the unlearned model's behavior across safety, general generation quality, and personalized fine-tuning effectiveness. Overall, a model with higher CLIP Score, Aesthetic Score, CLIP-I, CLIP-T, and DINO Score, and lower IP, US, and FID is favored, as it indicates better ability in generating fidelity-preserved, human preference-aligned general images, stronger personalized generation capability, and reduced risk of producing harmful or unsafe content.

## B.4 More Implementation Details

We used three datasets for training ResAlign, each serving a distinct objective. First, the harmful dataset $\mathcal{D}_{\text{harmful}}$ is used to compute the harmful loss $\mathcal{L}_{\text{harmful}}$. We constructed this dataset by selecting 150 unsafe prompts from existing unsafe datasets (e.g., NSFW-56k [43]). Second, the preservation dataset $\mathcal{D}_{\text{preserve}}$ is used to compute the regularization term $\mathcal{R}(\theta)$. For this dataset, we selected 140 benign prompts from COCO-Objects and CelebA-HQ. Finally, to simulate downstream fine-tuning data, we constructed the fine-tuning simulation dataset $\mathcal{D}_{\text{FT}}$ by randomly sampling 100 prompts from a prompt pool selected from DiffusionDB, NSFW-56k, and $\mathcal{D}_{\text{harmful}}$. For all datasets, we generated corresponding images for each prompt to form text–image pairs. We manually refined and reformatted some prompts to ensure consistency across all datasets. Notably, DiffusionDB is utilized in multiple contexts throughout our work. To avoid data leakage and overfitting, we manually verified that there is no overlap between any of the subsets used for different purposes. Additionally, gradient clipping and adaptive hyperparameter scheduling are also used to stabilize training.

For evaluation, unless otherwise stated, our fine-tuning is based on the official script provided by diffusers[13], whose default configuration is full-parameter fine-tuning on the UNet parameters with learning rate of $1 \times 10^{-5}$, a batch size of 1, and training step of 200, using AdamW [50] optimizer with default hyperparameters. For methods such as AdvUnlearn and Receler, where the unlearning is performed on modules (e.g., text encoder or auxiliary adapters) rather than on the UNet, we include these modules during downstream fine-tuning as well. This ensures a fair and comprehensive assessment of each method's worst-case resilience.

## C More Experimental Results

### C.1 Results on Adversarial Attacks & Potential Adaptive Attacks

While defending against adversarial attacks and adaptive attacks is not the primary focus of this work, we evaluate ResAlign's effectiveness when faced with several well-established attack strategies for unlearned diffusion models and explore a potential adaptive attack strategy to understand whether ResAlign can be easily bypassed or compromised with minimal effort.

---

[13]https://github.com/huggingface/diffusers/blob/main/examples/text_to_image

**Results on Existing Adversarial Attacks.** We first evaluate the robustness of ResAlign against several existing adversarial attacks, including SneakyPrompt [79], MMA-Diffusion [77], and Unlearn-DiffAtk [84], which is specifically designed against unlearned diffusion models. To comprehensively assess robustness under different threat models, we consider both black-box and white-box scenarios. In the black-box setting, we directly utilize the adversarial prompt benchmarks from SneakyPrompt and MMA-Diffusion. For each adversarial prompt, we generate 3 images using the

Tab. 10: Evaluation on existing adversarial attacks.

| Attack | ESD | AdvUnlearn | Ours |
|---|---|---|---|
| SneakyPrompt | 13.78% | 3.57% | 0.51% |
| MMA-Diffusion | 26.00% | 1.60% | 1.50% |
| UnlearnDiffAtk | 83.05% | 24.58% | 33.90% |

target model. A prompt is considered successful if at least one of the generated images is flagged as unsafe by the NudeNet detector. We follow prior work [77] and report the metric ASR-3, defined as the percentage of prompts for which at least one unsafe image is produced. In the white-box setting, we adopt the evaluation protocol of UnlearnDiffAtk [84], which involves using gradient information from the unlearned model to iteratively optimize adversarial prompts for maximum reactivation of harmful behaviors. This allows us to assess ResAlign's robustness under a strong, white-box adversarial setup, and we report the ASR metric, evaluated as the ratio of final successful optimized prompts as rated by NudeNet detector. As shown in Tab. 10, although ResAlign is not specifically designed to defend against adversarial prompts, it achieves a high level of robustness. Our method significantly outperforms ESD and is even comparable to AdvUnlearn, the state-of-the-art approach explicitly targeting adversarial prompt defense. Moreover, it is feasible to combine AdvUnlearn with our ResAlign unlearned model, which we verify can reduce the attack success rate of UnlearnDiffAtk to 8.47%. We will extend our method to defend against adversarial prompts in our future work.

**Discussion on Adaptive Attacks.** We further investigate whether ResAlign can be bypassed by adaptive strategies. Intuitively, ResAlign constrains the local loss landscape around the current unlearned parameters (see Eq. 5 and Proposition 1), which weakens harmful-gradient signals under fine-tuning. We thus evaluate two representative adaptive strategies that aim to escape this subspace.

First, we study a parameter-space escape attack, which augments the downstream fine-tuning objective with a term that actively pushes parameters away from the current solution: $\mathcal{L}_{\text{adaptive}}(\theta') = \mathcal{L}_{\text{FT}}(\theta'; \mathcal{D}_{\text{FT}}) - \lambda\|\theta' - \theta\|_p$, where we consider $p \in \{1, 2\}$ and set $\lambda = 0.5$. The intuition is that by encouraging the optimizer to move far from $\theta$, the model may escape the resilient region induced by ResAlign and thus recover harmful capability. However, as shown in Tab. 11, both variants fail to meaningfully bypass ResAlign, as their IP scores remain largely comparable to those obtained from standard fine-tuning.

Tab. 11: Results of the parameter-space escape attack. The reported metric is IP after fine-tuning.

| Setting | DreamBench++ | DiffusionDB | Harmful-Imgs |
|---|---|---|---|
| $p = 1$ | 0.0294 | 0.0523 | 0.2381 |
| $p = 2$ | 0.0283 | 0.0512 | 0.2402 |

Second, we evaluate a progressive relearning attack inspired by [17], which hypothesizes that gradually exposing the model from clean to increasingly unsafe data may reopen memory pathways and strengthen recovery. To simulate this, we split the 200-step fine-tuning into 4 equal stages: the first two use benign datasets (Dream-Bench++ and CelebA-HQ), the third stage uses a relatively safer subset of Harmful-Imgs (bottom 50% as ranked by NudeNet), and the final stage uses the full Harmful-Imgs set. As shown in Fig. 7, we observe that while this strategy brings better IP increase than single datasets, the IP does not substantially increase until the final stage where explicit harmful images are introduced, indicating the progressive schedule alone cannot meaningfully circumvent ResAlign without access to sufficient harmful data. Overall, we conclude that

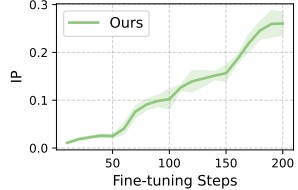

Fig. 7: Evaluation results of the progressive relearning attack.

it is non-trivial to effectively bypass ResAlign through simple adaptive modifications to the loss function or data exposure schedule in the absence of sufficient harmful data.

## C.2 Comparison with Additional Baselines.

To further strengthen our evaluation, we additionally compare ResAlign with 3 recent unlearning methods: RACE [33], RECE [18], and Receler [25]. As summarized in Tab. 12, while these methods achieve strong pre-fine-tuning safety performance, they consistently exhibit noticeable rebounds in toxicity after downstream fine-tuning on both DreamBench++ and DiffusionDB. Two key

observations can be drawn from these results. First, the post-fine-tuning safety degradation remains a universal issue across all evaluated methods, including these newly proposed ones, suggesting that current unlearning methods generally lack resilience under downstream fine-tuning. Second, stronger pre-fine-tuning safety does not necessarily imply better post-fine-tuning resilience. For example, RACE achieves low initial IP but suffers notable safety deterioration after fine-tuning. These findings further corroborate our central conclusion: existing unlearning approaches primarily optimize for immediate safety rather than long-term resilience, underscoring the importance of developing frameworks like ResAlign that explicitly address this gap.

Tab. 12: Comparison with additional baselines. We report the results averaged over three independent runs. The metric is IP.

| Method | No FT | DreamBench++ | DiffusionDB |
|--------|-------|--------------|-------------|
| RACE | 0.031 | 0.127 | 0.217 |
| RECE | 0.043 | 0.074 | 0.116 |
| Receler | 0.063 | 0.142 | 0.183 |

We hope these results can inspire future work toward more resilient safety alignment methods.

### C.3 More Ablation Study & Hyperparameter Analysis Results

**Effect of Meta Learning.** To further verify the effectiveness of our meta-learning design, we conduct an ablation study by varying the diversity of fine-tuning algorithms used during meta-training. Specifically, we train ResAlign with and without meta learning, yet with different parameter selection schemes. The results are summarized in Tab. 13. From these results, two insights can be drawn. First, without meta-learning, ResAlign tends to

Tab. 13: Effect of meta-learning. The metric is IP.

| Training Config | Full Param. | LoRA | LyCORIS | SVDiff |
|-----------------|-------------|------|---------|--------|
| Full Param. only (w/o ML) | 0.028 | 0.084 | 0.074 | 0.043 |
| LoRA only (w/o ML) | 0.096 | 0.045 | 0.069 | 0.047 |
| Full Param. + LoRA (w/ ML) | 0.030 | 0.051 | 0.054 | 0.036 |

overfit to the specific fine-tuning algorithm seen during training. For example, although LoRA and LyCORIS share similar structures, a model trained only on LoRA performs poorly when applied to LyCORIS, consistent with prior findings that single-configuration hypergradients can overfit to local settings [49, 44]. Second, incorporating multiple configurations during meta-learning (e.g., Full Param. + LoRA) not only improves in-distribution performance but also enhances cross-algorithm generalization, even to unseen fine-tuning methods such as SVDiff. These results confirm that meta-learning effectively mitigates overfitting and yields more generalizable hypergradients.

**Effect of $\beta$.** We further study the effect of the regularization weight $\beta$, which controls the relative strength of the resilience regularization term in ResAlign. As shown in Tab. 14, a very small $\beta$ leads to insufficient regularization, whereas an excessively large $\beta$ (close to 1.0) may cause the model to overlook current harmfulness signals and lead to unstable training. Nevertheless, ResAlign remains robust within a wide and practical range (e.g., $\beta \in [0.3, 0.9]$), consistently achieving strong safety performance on both DreamBench++ and DiffusionDB. This demonstrates that ResAlign is not overly sensitive to the choice of $\beta$.

Tab. 14: Effect of $\beta$. The metric is IP after fine-tuning.

| $\beta$ | 0.1 | 0.3 | 0.5 | 0.7 | 0.9 | 1.0 |
|---------|-----|-----|-----|-----|-----|-----|
| DreamBench++ | 0.094 | 0.038 | 0.046 | 0.018 | 0.017 | 0.160 |
| DiffusionDB | 0.181 | 0.096 | 0.069 | 0.083 | 0.061 | 0.241 |

**Effect of $\gamma$ on DiffusionDB dataset.** In the main paper, we evaluated the impact of different values of $\gamma$ on our ResAlign's performance using the DreamBench++ dataset. In this section, we also conduct the ablation study on the DiffusionDB dataset. As shown in Fig. 8, a similar trend is observed on the DiffusionDB dataset. This supports our conclusion that ResAlign achieves stable resilience to downstream fine-tuning across a wide range of $\gamma$ values, indicating that ResAlign is not overly sensitive to the choice of this hyperparameter.

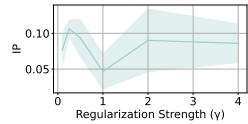

Fig. 8: Effect of $\gamma$.

### C.4 Computational Efficiency

We further compare the computational cost of ResAlign with representative baselines in terms of both training time and average GPU memory usage. As shown in Tab. 15, our method achieves a favorable balance between efficiency and performance. Specifically, ResAlign requires comparable average memory to most full-parameter fine-tuning baselines (e.g., LCFDSD-NG). Notably, although our peak GPU memory

Tab. 15: Comparison of computational cost.

| Method | Avg. VRAM (GB) | Training Time (min) |
|--------|----------------|---------------------|
| ESD | 12.4 | 14.3 |
| AdvUnlearn | 29.4 | 210.3 |
| LCFDSD-NG | 20.3 | 38.4 |
| Meta-Unlearning | 35.7 | 725.4 |
| ResAlign (Ours) | 25.4 | 58.2 |

usage temporarily doubles (compared to average GPU memory usage) during Hessian-vector product computation, this overhead occurs only briefly. Overall, ResAlign remains computationally practical and well-suited for real-world unlearning applications.

# D More Related Work

## D.1 Flat-Minima Optimization and Hessian Trace Regularization

As discussed in Proposition 1, our ResAlign implicitly imposes a regularization on the trace of the Hessian of the harmful loss, possibly encouraging the model to converge to flatter regions of the loss landscape. This connects our work to the broader literature on Hessian-based regularization and flat-minima optimization, which has been broadly studied in optimization and generalization theories.

Classical approaches such as Hutchinson's estimator [29], Lanczos approximation [72], and Chebyshev polynomial methods [9] have been proposed to approximate or minimize the trace of the Hessian. These techniques have found applications in diverse areas including convex optimization and physical simulation. However, they typically rely on stochastic or numerical estimators that are either non-differentiable or computationally intensive to differentiate through, making direct optimization of the Hessian trace largely impractical for modern large-scale neural networks.

In deep learning, several works have explored indirect flatness optimization, such as Sharpness-Aware Minimization [11] and Entropy-SGD [2], which aim to improve generalization by favoring flat minima in the loss landscape. Yet, they are primarily designed for classification tasks and have not been extensively studied for diffusion models or investigated from the perspective of safety resilience.

Our work differs in both motivation and mechanism. Rather than explicitly minimizing the Hessian trace or relying on costly curvature estimation, ResAlign achieves a similar flatness-inducing effect implicitly through a simple resilient unlearning objective. This design efficiently regularizes the harmful loss landscape toward flatter regions without introducing direct Hessian computations, making it particularly well-suited for large-scale diffusion models and safety-driven unlearning.

## D.2 Meta Learning and Bi-level Optimization

Meta-learning, or "learning to learn," is a classical paradigm in machine learning that aims to train a meta-model capable of rapidly adapting to new tasks with limited data [10, 63]. The high-level idea is to learn across multiple tasks such that the resulting meta-parameters encode general learning dynamics that transfer efficiently to unseen scenarios. Unlike conventional meta-learning, whose primary goal is to improve few-shot adaptation, our work leverages meta-learning to enhance the generalizability of hypergradient estimation and to mitigate overfitting to a specific simulated fine-tuning configuration.

Meta-learning can also be formulated as a bi-level optimization problem, where the outer optimization updates the meta-parameters to achieve better generalization across tasks, and the inner optimization represents task-specific adaptation. From this perspective, our framework is closely related to bi-level optimization and its variants that estimate the hypergradient (or "meta-gradient") connecting the two levels. A central challenge in such formulations lies in efficiently computing this hypergradient. To address this, prior works have developed approximate implicit differentiation (AID) methods that estimate gradients through the optimality condition of the inner problem [19, 31]. Our method can be viewed as a special instance of AID derived under the Moreau envelope-based reformulation, which enables efficient hypergradient estimation without unrolling fine-tuning trajectories.

As acknowledged in Appendix A, iMAML [63] is, to the best of our knowledge, the most technically related prior work to ours, which also introduces a proximity term in the inner optimization to regularize the update w.r.t. the base parameters. However, our work differs in two key aspects. First, the optimization objective fundamentally diverges: iMAML aims to improve few-shot generalization, whereas we aim to mitigate the recovery of harmful behaviors during downstream fine-tuning. Second, iMAML requires explicitly enforcing the proximity term during the inner optimization of each fine-tuning task, which inherently constrains the optimization dynamics of the lower-level task. In contrast, our method does not require any modification to the downstream fine-tuning loss. Instead, we treat the fine-tuned model as the minimizer of an ME objective when estimating the hypergradient. This design, facilitated by the properties of diffusion models (see more analyses in Appendix E.1), allows

ResAlign to impose no constraints on the choice of downstream loss, making it broadly applicable. Beyond these key aspects, our work also differs in the technique for solving the linear system, and has proposed a cross-configuration meta learning technique.

### D.3 Defending against Harmful Fine-tuning

The degradation of safety alignment after downstream fine-tuning has recently emerged as an important and actively studied phenomenon. Most existing research [26, 23, 68, 73, 28, 27] focuses on large language models, where researchers have explored various strategies to mitigate such degradation, including interventions during pretraining, safety-aligned fine-tuning, and controlled data filtering or objective design. Among them, Booster [28] is most relevant to our work, which works by estimating the effect of a single-step harmful fine-tuning update on model safety and employs a first-order approximation to compute a defensive gradient update.

In contrast, research on defending against harmful fine-tuning in diffusion models remains largely underexplored. To the best of our knowledge, besides the methods discussed in Sec. 2 (e.g., LCFDSD), the concurrent work Meta-Unlearning [15] is also directly related to our scenario. Additionally, while works such as IMMA [85] and SOPHON [8] were originally developed for preventing diffusion models from learning specific tasks or concepts, their underlying mechanisms can also be adapted to enhance resilience against safety degradation after fine-tuning.

The key distinction between our approach and these existing methods lies in how the hypergradient is estimated and how the downstream fine-tuning configurations are selected. Specifically, our framework introduces (i) a principled implicit differentiation based on the Moreau envelope approximation, and (ii) cross-configuration generalization via meta-learning. These two components jointly enable ResAlign to more accurately capture the higher-order interactions between model parameters and downstream fine-tuning dynamics, leading to improved resilience against harmful behavior recovery. We provide further discussion and experimental comparisons in Appendix E.1 and Sec. 5.

## E  More Discussion

### E.1  Discussion on Other Approximations

In this section, we discuss our choice of the Moreau envelope–based approximation for estimating the hypergradient. In our early trials, we also explored several other viable alternatives. The first is the first-order approximation widely adopted by previous works [85, 28, 15, 8], which directly assumes the Jacobian satisfies $\frac{\partial \theta_{\text{FT}}^*}{\partial \theta} \approx I$, thus simplifying the hypergradient as $\nabla_\theta \mathcal{L}_{\text{harmful}}(\theta_{\text{FT}}^*) \approx \nabla_{\theta_{\text{FT}}^*} \mathcal{L}_{\text{harmful}}(\theta_{\text{FT}}^*)$. Although computationally efficient, this approach yields substantially degraded performance compared to our ME method (IP: 0.16 vs. 0.07 on DiffusionDB), as it neglects the higher-order interactions between base and fine-tuned parameters. Besides, we also considered unrolled differentiation [51], which computes the hypergradient by backpropagating through the full fine-tuning trajectory. While theoretically accurate, it is computationally prohibitive for large diffusion models. For instance, on SD v1.4, a single A100 GPU can only unroll up to 3 steps, far too few to capture meaningful fine-tuning dynamics. Finally, our ME-based approximation models the fine-tuned parameters as the minimizer of a ME objective and applies implicit differentiation to estimate the hypergradient by solving a linear system via Richardson iteration. This approach only requires access to the final fine-tuned model, making it configuration-agnostic and highly scalable. Empirically, it is nearly as fast as the first-order method (approximately 58 min vs. 50 min) but delivers much better alignment performance. We thus finally adopt ME-based approximation.

One key underlying insight behind our ME-based approximation is that pretrained diffusion models typically require only small parameter updates to adapt to downstream tasks [36], and in some cases (e.g., Textual Inversion [12]), updating even a single embedding can suffice. Hence, viewing the fine-tuned model as the minimizer of a task-specific objective regularized by proximity to the base model aligns closely with the real fine-tuning behavior of modern diffusion models. This makes our approach both practical and theoretically well-suited for this defense scenario.

## E.2 Discussion on the Diversity of Meta Learning Configurations

A key principle underlying our design is to maintain both the dataset and configuration pool as diverse and representative as possible, which is crucial for the meta-learned update to generalize to unseen fine-tuning scenarios. This motivates us to propose the meta-learning technique. Currently, our implementation of ResAlign serves as a prototype to validate the effectiveness of the proposed meta-learning mechanism, and we adopt several representative fine-tuning configurations widely used in the diffusion community, such as standard LoRA-based and DreamBooth-style setups. This already provides strong generalization across real-world downstream tasks without evident overfitting (e.g., anime style transfer, object and face personalization, and art style learning, as shown in Tab. 1-4). Moreover, our framework is flexible-users can readily customize the dataset and configuration pool to better match their target domain. For instance, in avatar personalization tasks that primarily involve human faces and small learning rates, incorporating more face-related data and sampling smaller learning-rate configurations during meta-learning would likely yield better results.

To validate whether enriching configurations enhance our generalization, we have expanded our configuration pool to include a broader range of fine-tuning algorithms (e.g., LoRA with varying hyperparameters, LyCORIS, and QLoRA) and objective functions (e.g., SVDiff and CustomDiffusion losses). This integration is straightforward since our ME-based estimation only requires the final fine-tuned model, imposing no assumptions on the underlying optimization or configuration. Preliminary results indicate that this extension can further improve robustness—for instance, on DoRA+DiffusionDB, the updated model achieves an improved post-fine-tuning IP of 0.048 compared to 0.069 in the original setup. Future work can further expand the meta-learning pool and systematically investigate the interactions among different fine-tuning configurations, which may lead to even more effective and broadly generalizable meta-learned updates in ResAlign.

## E.3 Discussion on ResAlign's Effectiveness on Contaminated Data

Recall that our experiments in Fig. 4 show that ResAlign can also mitigate harmfulness rebound even when exposed to contaminated or harmful data during downstream fine-tuning. This phenomenon can possibly be theoretically explained by Proposition 1, which shows that ResAlign effectively imposes a penalty on the trace of the Hessian (i.e., the second-order derivatives) of the harmful loss with respect to model parameters. Intuitively, this encourages the model to converge to a locally flatter region of the loss landscape for the harmful objective. Ideally, around the optimized parameters, the first-order gradients of the harmful loss would be small in most directions. Consequently, even when the model encounters harmful samples, the gradient signals that could re-enable harmful behaviors remain weak, making it difficult for downstream fine-tuning to substantially increase harmfulness. In practice, achieving a perfectly flat region where all harmful gradients vanish is largely unrealistic due to the stochasticity of diffusion training and the high-dimensional landscape. Nonetheless, ResAlign empirically suppresses the harmful recovery to some extent, as reflected by the consistently lower IP observed even under contaminated conditions.

## E.4 On Instance-level and Concept-level Unlearning

Our work focuses on safety-driven unlearning, which aims to reduce model toxicity and unsafe generations. In the existing literature, such unlearning can be achieved through two complementary paradigms: (i) instance-wise unlearning [56, 43], which directly unlearns certain unsafe instances from the model (e.g., through gradient ascent or reinforcement learning-based preference optimization on a collected dataset), and (ii) concept-level unlearning [13, 14], which suppresses specific semantic concepts (e.g., "nudity" or "violence") by aligning or perturbing their corresponding representations. While these two strategies differ in implementation, both have been shown to effectively mitigate harmful behaviors and can generalize to unseen unsafe prompts. For example, our evaluation metrics (i.e., IP and US) are measured on prompts completely disjoint from the training set, yet both instance-level and concept-level approaches substantially reduce unsafe generations, suggesting that instance-level unlearning can also imply broader concept-level mitigation.

Importantly, our proposed ResAlign framework is loss-agnostic and can seamlessly incorporate either instance-level or concept-level unlearning objectives. As reported in Tab. 9, applying ResAlign on concept-level losses can also improve their post-fine-tuning safety, demonstrating that our framework generalizes across unlearning paradigms. Moreover, our experiments reveal that all existing unlearn-

ing baselines suffer from a significant safety drop after downstream fine-tuning, regardless of the paradigm of their unlearning objectives. This observation highlights a fundamental and underexplored challenge in safety-driven unlearning, i.e., the lack of resilience to downstream adaptation, which ResAlign takes an early yet meaningful step toward addressing.

### E.5 Discussion on Side Effects and Artifact Patterns

While ResAlign achieves strong safety resilience, we also observe several characteristic failure patterns associated with the use of the GA loss for unlearning. Specifically, GA may induce *over-rejection* behaviors, where the model rejects prompts that are safe in intent but semantically similar to unsafe ones seen during unlearning. Importantly, this phenomenon is not necessarily detrimental. In some cases, even semantically safe prompts can inadvertently yield unsafe generations, and such conservative refusal can effectively prevent these failures from occurring. Besides possible over-rejection, during repeated experiments, we find that even under identical configurations, the rejection patterns may vary across training runs. As shown in Fig. 9, besides the global wavy or scale-like mosaic artifacts across the entire image as shown in Fig. 1, typical artifact patterns also include (but are not limited to): (a)–(b) muscle-like distortions on human bodies, (c)–(d) incomplete global denoising, (e)–(f) wave- or grid-shaped destructive streaks, (g)–(h) localized wavy or scale-like mosaic patterns on human bodies. While these artifacts effectively suppress unsafe content, we acknowledge that some of them may negatively impact user

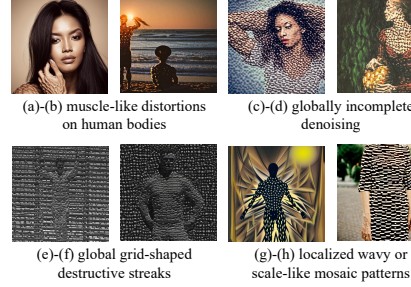

(a)-(b) muscle-like distortions on human bodies    (c)-(d) globally incomplete denoising

(e)-(f) global grid-shaped destructive streaks    (g)-(h) localized wavy or scale-like mosaic patterns

Fig. 9: Visualization results on rejection artifact patterns observed in our experiments.

experience or even cause discomfort (e.g., for individuals with trypophobia). Thus, we strongly encourage practitioners to consider this side effect brought by GA, and carefully evaluate model behaviors before real-world deployment. User discretion is also recommended.

For users who prefer safer yet more visually coherent outputs, substituting the GA-based loss with a milder unlearning objective (e.g., ESD's loss) can alleviate such over-rejection behaviors, as even rejected attempts will be given a safe image instead. We encourage future research to systematically study these patterns and develop better loss designs that balance safety with user experience.

### E.6 Limitations & Future Work

Our work still has the following limitations, which we aim to address in future work. First, our approach requires simulating fine-tuning during the unlearning process, which introduces additional computational overhead. However, the total cost remains within a practical range (less than one GPU hour for a full run in our main experiments), which is significantly more efficient than retraining a model from scratch. In real-world scenarios, users can choose methods that best balance their needs for resilience and safety given their computational budget. Second, our Moreau envelope-based approximation is inherently an approximation. Nevertheless, our experiments show that it is robust to long fine-tuning steps and a wide range of learning rates, and it outperforms methods based on first-order approximations. As discussed in the main paper, obtaining fully accurate hypergradients would require tracking and computing full Hessian matrix products, which is computationally prohibitive for diffusion models. Future work could explore more accurate yet efficient estimation strategies. Third, we acknowledge that when the downstream fine-tuning dataset is entirely composed of toxic data, both our method and existing baselines inevitably experience an increase in harmfulness. We believe that perfect resilience under such adversarial conditions is intrinsically difficult, given the strong generalization and few-shot capabilities of modern diffusion models. Nonetheless, our method demonstrates stronger resilience compared to baselines, maintaining relatively low IP even under fully toxic fine-tuning. Fourth, similar to previous works, the NSFW classifiers used in our experiments are not perfectly accurate, and there might be false negatives or false positives. However, as all methods are evaluated under the same setting, the results are still fair and comparable to a large extent. We hope future work can build more accurate classification models. Finally, while we designed and evaluated adversarial & adaptive attacks and demonstrated ResAlign's robustness against them, we recognize that stronger or more sophisticated attacks may emerge in the future. Developing robust defenses against such potential threats remains an important direction for future work.

## E.7 Ethics Statement & Broader Impact

This work aims to address the safety challenges of text-to-image diffusion models, particularly their vulnerability to unsafe behavior inherited from toxic pretraining data and re-emerging during downstream fine-tuning. Our proposed method, ResAlign, is designed specifically to improve the resilience of safety-driven unlearning techniques, helping to mitigate the unintended recovery of harmful behaviors when models are fine-tuned on downstream data. We have carefully reviewed and ensured that our research adheres to the NeurIPS Ethics Guidelines for Authors[14]. All experiments are conducted using publicly available datasets and established benchmarks under their original licenses, or are constructed from filtered or recombined subsets of these datasets, ensuring no new unsafe content is introduced. All illustrated potentially sensitive outputs in this paper have sensitive regions carefully masked for viewer protection. We do not release any attack-related tools or data; any derived resources will be made available through gated access upon request. We will responsibly notify relevant developers if future findings reveal potential vulnerabilities. This work is intended solely for defensive and safety-enhancing purposes; we do not endorse or support any offensive or unethical applications. We believe that our contributions will help advance the trustworthy and responsible development of generative AI technologies.

---

[14]https://neurips.cc/public/EthicsGuidelines

