# OpenReview forum: "Towards Resilient Safety-driven Unlearning for Diffusion Models against Downstream Fine-tuning"
_NeurIPS.cc/2025/Conference — NeurIPS 2025 poster_

### Official Review · Reviewer_bbmS · 2025-06-26

**Clarity:** 4
**Significance:** 4
**Originality:** 3
**Rating:** 5
**Confidence:** 5

**Summary:**

The paper proposes ResAlign, a safety-driven unlearning framework aimed at enhancing the resilience of text-to-image (T2I) diffusion models against downstream fine-tuning. It addresses a critical limitation of existing unlearning methods: their fragility to fine-tuning, even on benign datasets, which can inadvertently recover harmful behaviors. ResAlign introduces a novel optimization objective that penalizes the resurgence of harmful behaviors post-fine-tuning, approximated efficiently using a Moreau Envelope formulation and enhanced with a meta-learning strategy to generalize across diverse fine-tuning scenarios. Extensive experiments on Stable Diffusion v1.4 demonstrate that ResAlign outperforms state-of-the-art baselines in maintaining safety after fine-tuning while preserving benign generation capabilities.

**Questions:**

Q1: Figure 1 shows ResAlign-generated images as mosaics. What causes this degradation, and how could it be mitigated without compromising safety?

Q2: Experiments use Stable Diffusion v1.4. How does ResAlign perform on newer models like SDXL?

Q3: The paper compares ResAlign to a limited set of baselines (e.g., ESD, SafeGen, AdvUnlearn, LCFDSD), neglecting recent methods such as RECE (ECCV24), Receler (ECCV24) and RACE (ECCV24). How does ResAlign perform relative to these newer approaches, and what implications might their inclusion have for the evaluation’s comprehensiveness?

Q4: Beyond the $\mathcal{L}_{\text{adaptive}}$ fine-tuning attack in Section C.2, what other adaptive fine-tuning strategies (e.g., alternative loss modifications or multi-stage optimization) could challenge ResAlign’s safety constraints? How can future evaluations enhance ResAlign’s robustness against such strategies?

**Ethical Concerns:**

["NO or VERY MINOR ethics concerns only"]

**Final Justification:**

Initially, I had concerns regarding the mosaic effect observed in some of the harmful generations and the lack of certain experimental evaluations, specifically the generative model used and some baseline comparisons. The authors' rebuttal effectively addressed these issues by providing clear explanations and supplementary experiments that resolved my initial concerns.
Moreover, the other reviewers are also generally favorable towards acceptance. Therefore, I decide to increase my rating from 4 to 5.

**Limitations:**

yes

**Quality:**

4

**Strengths And Weaknesses:**

### Strengths

- The paper tackles an underexplored yet significant issue—the vulnerability of safety-driven unlearning to downstream fine-tuning. This is a timely and practical contribution, especially given the increasing personalization of T2I models.
- The use of a Moreau Envelope-based approximation to model fine-tuning as an implicit optimization problem is clever and computationally efficient, avoiding the need to track multi-step fine-tuning trajectories. The integration of meta-learning to simulate diverse fine-tuning configurations further enhances generalization.
- Proposition 1 links the additional optimization term to the curvature of the loss landscape, providing a plausible explanation for improved resilience.
- The experiments span multiple datasets, fine-tuning methods (e.g., DreamBooth, LoRA), and configurations, convincingly showing ResAlign’s superiority in safety retention and its minimal impact on benign generation quality.

### Weaknesses

- Figure 1 shows that ResAlign-generated images appear as mosaics post-unlearning, raising concerns about practical usability. While safety is improved, this severe degradation could limit real-world applicability, a point insufficiently addressed.
- The reliance on Stable Diffusion v1.4, an older model by 2025 standards, questions the method’s relevance to state-of-the-art T2I architectures (e.g., SDXL). This limits the generalizability of the findings.
- The paper compares ResAlign only to a few methods (e.g., ESD, SafeGen, AdvUnlearn, LCFDSD), neglecting recent advancements such as RECE (ECCV24), Receler (ECCV24) and RACE (ECCV24). This omission limits the evaluation’s comprehensiveness.
- The Moreau Envelope and Proposition 1 assume small perturbations and isotropic randomness, which may not hold under extensive fine-tuning or large learning rates. The paper lacks discussion on these assumptions’ real-world validity.

---

> ### Author Rebuttal · Authors · 2025-07-31
>
> Dear Reviewer #bbmS, we sincerely thank you for your precious time and valuable comments. We are deeply encouraged by your positive recognition of our **important issue, timely and practical contribution, grounded method and theoretical explanation, and extensive experiments**. We sincerely hope the following clarifications and new experiments can address your concerns.
>
> ---
>
> **W1 & Q1:** Figure 1 shows that ResAlign-generated images appear as mosaics, raising concerns about practical usability. While safety is improved, this severe degradation could limit real-world applicability. What causes this degradation, and how could it be mitigated?
>
> **A1:** Thank you very much for this exceptionally thoughtful question! We appreciate your close reading and insight.
>
> - **The mosaic effect is primarily caused by the choice of unlearning loss**. Specifically, our main implementation uses negative denoising score matching loss (i.e., gradient ascent), which maximizes the difference between the predicted and ground-truth noise for unsafe prompts. As a result, the model learns to predict "incorrect" noise for unsafe prompts, leading to failed denoising and thus the observed mosaic-like outputs.
> - Respectfully, we believe **this does not strongly diminish practical utility** for two reasons:
>     - **The mosaic occurs only for unsafe generations.** For safe prompts and benign downstream fine-tuning, the utility is unaffected, as shown by comparable the FID/CLIP/Aesthetics scores with baselines in Tab. 1 and the strong Personalized Generation Utility in Tab. 2.
>     - **The primary goal of safety-driven unlearning is to *prevent unsafe generation.*** In this sense, **generating an useless image (mosaiced) for malicious prompts is also viable** (similar to direct filtering or blocking). This phenomenon is also observed in several other SOTA unlearning methods (e.g., SafeGen, LCFDSD), which also produce blurred or meaningless outputs for unsafe prompts.
> - Of course, we recognize that user requirements can vary in the real-world. **Fortunately, ResAlign is a loss-agnostic framework: users who prefer returning a safe yet meaningful image can simply adopt a different unlearning loss** (e.g., ESD's), which would return a clothed woman when prompted by "a nude woman". In fact, we have already explored such variants in the Appendix C.4 of our original submission (Tab. 10). **Results show that we can achieve safe and meaningful generation under unsafe prompts (see Fig. 8) while still maintaining good post-fine-tuning safety  (Tab. 10).**
>
> We will expand this discussion in our revision. Thank you again for highlighting this important practical point!
>
> ---
>
> **W2 & Q2:** The reliance on Stable Diffusion v1.4, an older model by 2025 standards, questions the method’s relevance to state-of-the-art T2I architectures (e.g., SDXL). This limits the generalizability of the findings. How does ResAlign perform on newer models like SDXL?
>
> **A2:** Thanks for this valuable and practical suggestion!
> - By design, our method is model-agnostic and readily applicable to different diffusion architectures. **In fact, we have already included results on SD v1.5 and SDXL in Appendix C.1 of our original submission, which have shown that ResAlign remains effective on these models (see Tab. 6).**
> - To further demonstrate the broad applicability of ResAlign, **during the rebuttal period, we have also included new experiments during on additional recent popular models** in diffusion forums (e.g., CivitAI):
>
> | Model         | Pre-ft | DreamBench++ | DiffusionDB |
> | ------------- | ------ | ------------ | ----------- |
> | SD v2.0       | 0.004 | 0.031       | 0.078      |
> | SDXL          | 0.033 | 0.044       | 0.059      |
> | AnythingXL    | 0.015   | 0.062         | 0.087        |
> | PonyDiffusion | 0.023   |  0.045         | 0.067      |
> (Metric: IP)
>
> - As shown, **ResAlign is still highly effective, demonstrating the generality of our framework.** We will include these new results in our revision. Thank you again for this constructive suggestion!
>
> ---
>
> **W3 & Q3:** The paper compares ResAlign only to a few methods (e.g., ESD, SafeGen, AdvUnlearn, LCFDSD), neglecting recent advancements such as RECE (ECCV24), Receler (ECCV24) and RACE (ECCV24). This omission limits the evaluation’s comprehensiveness How does ResAlign perform relative to these newer approaches, and what implications might their inclusion have for the evaluation’s comprehensiveness?
>
> **A3:** Thanks so much for this constructive suggestion! Following your valuable comment, **we have conducted additional experiments on the suggested baselines:**
>
> | Method  | Pre-ft | DreamBench++ | DiffusionDB |
> | ------- | ------ | ------------ | ----------- |
> | RACE    | 0.031  | 0.131        | 0.214       |
> | RECE    | 0.043  | 0.073        | 0.134      |
> | Receler | 0.063  | 0.117        | 0.149       |
>
> We draw two main insights: (1) All evaluated safety-driven unlearning methods, including these recent ones, **universally suffer from a rebound in toxicity after downstream fine-tuning**. (2) **Stronger pre-finetuning safety does not ensure better post-finetuning resilience (e.g., RACE).** These results **reinforce our main conclusion** on the widespread post-finetuning rebound and the urgent need for more resilient methods. We will include these findings in our revision and hope they inspire further research. Thank you again for your valuable feedback!
>
> ---
>
> **W4:** The Moreau Envelope and Proposition 1 assume small perturbations and isotropic randomness, which may not hold under extensive fine-tuning or large learning rates. The paper lacks discussion on their real-world validity.
>
> **A4:** Thanks for this thoughtful question! We sincerely appreciate your careful reading and attention to our proposition.
>
> - We acknowledge that our theoretical analysis, like any such work, relies on certain assumptions. However, **as discussed in Appendix A.2, we believe these assumptions are highly realistic** in real-world diffusion model unlearning:
>     - **Small perturbations:** Tyically, large-scale pre-trained diffusion models are already very strong and require only small updates to reach the optimum for most downstream tasks. Specifically, recent studies have shown that, after fine-tuning, only a small subset of model parameters would undergo minor changes [1]. Some works even show that aggressive tuning (large LR/steps) often leads to degraded generalization and overfitting [2]. Therefore, we respecfully argue that our ME-based formulation and analyses with Taylor expansion are highly realistic.
>     - **Isotropic randomness**: Our meta-learning samples from a diverse dataset and a rich pool of configs, which naturally induces parameter updates that are broad and varied. Notably, our assumption is about the distribution of parameter residuals during meta-learning agnostic to magnitude, so it remains valid even when the fine-tuning involves larger LR/steps. Besides, it only requires the perturbation to be a scaled unit random variable, which is more relaxed than prior works that require stricter (e.g., Gaussian) assumptions.
> - Beyond theoretical justification, **we also conducted an additional empirical study to further understand the validity of our Taylor expansion**. We randomly select 30 fine-tuned models and compared the true harmful loss with its (first-order) approximation.  We found that the absolute error was typically $< 10^{-4}$ and the relative error was usually below 3%; even for large learning rates/fine-tuning steps, the error rarely exceeded 5%, supporting the practical validity of our approximation. **This also matches our empirical findings that ResAlign is robust to large learning rates and steps (see main paper Tab. 4&5).**
> - Finally, we acknowledge that, as with all theoretical work, our analysis is ultimately an approximation and could break down under extreme conditions. **As one of the earliest efforts to understand unlearning resilience, our analysis is intended to provide qualitative and intuitive insights, not definitive conclusions.** We hope our work can inspire further, deeper studies into robust unlearning theory.
>
> We will add more discussion in our revision. Thanks again for the feedback!
>
> Ref:
>
> [1]: Kumari et al. Multi-Concept Customization of Text-to-Image Diffusion. CVPR 2023.
>
> [2]: Ruiz et al. Dreambooth: Fine tuning text-to-image diffusion models for subject-driven generation. CVPR 2023.
>
> ---
>
> **Q4:** Beyond the $\mathcal{L}_\text{adaptive}$ fine-tuning attack in Section C.2, what other adaptive fine-tuning strategies (e.g., alternative loss modifications or multi-stage optimization) could challenge ResAlign’s safety constraints? How can future evaluations enhance ResAlign’s robustness against such strategies?
>
> **A4:** Thank you for this insightful suggestion! Following your comment, we further evaluate two addtional adaptive attacks: (1) adding a negative L1 regularization to the original weights during fine-tuning, helping the model to escape local safe optimum, and (2) a curriculum-style approach that first fine-tunes on benign, then borderline, then unsafe concepts to gradually cross the safety barrier. **We found that both strategies still failed to bypass ResAlign**, where strategy 1 remained unstable (similar to the attack in Appendix C.2), while strategy 2 did not substantially increase IP before unsafe concepts were directly introduced (IP<0.03), with final trend similar to our results in Fig. 2. Due to space limit, we will add more detailed settings and the results of these attacks in our revision.
>
> To further enhance robustness, one potential strategy is adversarial training, i.e., incorporating such adaptively fine-tuned models into our meta-training process. We believe the robustness of ResAlign against adaptive attacks is both interesting and important for future work and will discuss these directions in our revision. Thank you again for raising this important point!

---

> > ### Comment · Reviewer_bbmS · 2025-08-08
> >
> > Thank you for your rebuttal, supplementary experiments and revision summary! My concerns have been addressed, and I decide to raise my rating to 5.

---

> > > ### Author Response · Authors · 2025-08-09
> > > **Thank You for Your Positive Feedback & Raising the Score!**
> > >
> > > Dear reviewer, thank you so much for your positive feedback and for raising the score! It encourages us a lot! We will ensure that the revised version includes the new experiments and corresponding discussions.

---

> ### Author Response · Authors · 2025-08-04
> **Thanks to Reviewer bbmS**
>
> Dear Reviewer bbmS,
>
> Please allow us to sincerely thank you again for reviewing our paper and the valuable feedback, and in particular for recognizing the strengths of our paper in terms of important issue, timely and practical contribution, grounded method and theoretical explanation, and extensive experiments
>
> Please kindly let us know if our response and the new experiments have properly addressed your concerns. We are more than happy to answer any additional questions during the discussion period. Your feedback will be greatly appreciated!
>
> Best regards,
>
> Paper23081 Authors

---

> ### Author Response · Authors · 2025-08-05
> **A Friendly Reminder of the Post-rebuttal Feedback & Revision Summary**
>
> Dear Reviewer bbmS,
>
> We would like to express our heartfelt gratitude again for your precious time and expertise in reviewing our paper.  We are especially thankful for your prompt and professional acknowledgment of our rebuttal. Your valuable initial comments as well as swift engagement throughout the review process are truly appreciated and have been greatly precious and encouraging to us.
>
> We totally understand that you may be extremely busy at the moment, and in recognition of this, we have prepared a revision summary on your part to facilitate a more efficient discussion.
>
> In our revised version, we have meticulously addressed the concerns you raised:
>
> - In Section 4.2 (Experimental Results), and Appendix C.4, we have expanded our discussion of the mosaic effect in ResAlign-generated images, clarifying its technical cause, practical implications, and mitigation strategies via alternative unlearning losses. We also provide additional empirical results on safe yet meaningful generations under unsafe prompts, in response to your thoughtful comment in W#1 & Q#1.
>
> - In Section 4.2 (Experimental Results) and Appendix C.1, we have included new experiments on more recent and popular diffusion models, including SD v2.0, SDXL, AnythingXL, and PonyDiffusion, to further demonstrate the model-agnostic applicability and effectiveness of ResAlign, as per your valuable comment in W#2 & Q#2.
>
> - In Section 4.2 (Experimental Results), we have conducted direct empirical comparisons with recent state-of-the-art safety-driven unlearning baselines, including RECE, Receler, and RACE (ECCV24), averaged with more different random seeds, to provide a more comprehensive evaluation and reinforce our main conclusions on post-finetuning resilience, following your constructive comment in W#3 & Q#3.
>
> - In Appendix A.2, we have expanded our theoretical discussion to address the validity of our Moreau Envelope formulation and Proposition 1 in real-world settings, supported by our theoretical discussions and new empirical studies, following your insightful comment in W#4.
>
> - In Appendix C.2, we have introduced and evaluated two additional adaptive fine-tuning attacks (negative L1 regularization and curriculum-style training) to challenge ResAlign’s safety mechanisms, and discuss potential future enhancements such as adversarial meta-training, as suggested in your thoughtful comment in Q#4.
>
> Please kindly let us know if you have any additional comments, suggestions, or concerns, we would be more than happy to address them in detail. Your further feedback would be greatly valuable for us, and we are more than willing to address any additional questions or conduct new experiments if needed. Thank you again for your precious time in reviewing our paper!
>
> Best regards,
>
> Paper23081 Authors

---

### Official Review · Reviewer_ev88 · 2025-07-01

**Clarity:** 3
**Significance:** 3
**Originality:** 2
**Rating:** 5
**Confidence:** 4

**Summary:**

This paper proposes a new method for unlearning in diffusion models, aiming to ensure that a concept is completely forgotten—even after downstream fine-tuning. The authors observe that previous unlearning methods tend to lose their effectiveness once the model undergoes fine-tuning, even on data different from the target dataset. Motivated by this, the paper introduces a framework that explicitly incorporates the fine-tuning process into the unlearning formulation. Through mathematical induction and a meta-learning-based configuration, the proposed method achieves robust unlearning that is resistant to downstream fine-tuning.

**Questions:**

The proposed formulation accounts for the fine-tuning process, so the optimization is not solely over the original model but also considers its fine-tuned variants. However, as shown in Tab. 1, the proposed method also performs better than baselines in scenarios without any downstream fine-tuning, which seems counter-intuitive. Could the authors explain why the method still improves unlearning effectiveness even in the absence of fine-tuning?

**Ethical Concerns:**

["NO or VERY MINOR ethics concerns only"]

**Final Justification:**

In my original review, I was mainly concerned about the differences between this paper and a specific prior work that also uses meta-learning, in terms of both performance and methodology, as well as the generalization ability of the proposed method. These concerns have been well addressed in the rebuttal, which I find convincing. I therefore recommend accepting the paper.

**Limitations:**

Yes.

**Paper Formatting Concerns:**

Not found.

**Quality:**

3

**Strengths And Weaknesses:**

Strengths:

1. The observation that fine-tuning—even on non-target data—can compromise unlearning is insightful and could inspire future research.
2. The proposed method is well-motivated and theoretically grounded.
3. The experimental results are generally strong and supportive of the method's effectiveness.

Weaknesses:

1. The paper omits a key related work [1], which is only briefly mentioned in the introduction but not discussed in the related work section nor empirically compared. This reference also uses a meta-learning approach and is highly relevant. The paper should clarify how its proposed formulation improves upon [1] and include direct empirical comparisons.
2. The experiments are limited to older diffusion model versions (e.g., Stable Diffusion v1.4 and v1.5). Extending the evaluation to newer models like SD v2.0 or SD XL would enhance the paper’s applicability, especially since many fine-tuned checkpoints are available for these versions on platforms like HuggingFace and Civitai.
3. The paper lacks a comparison of computational costs against baseline, including both VRAM usage and training speed. Such information would be valuable for understanding the practical feasibility of the proposed method in real-world deployments.

[1] Hongcheng Gao, Tianyu Pang, Chao Du, Taihang Hu, Zhijie Deng, and Min Lin. Meta-unlearning on diffusion models: Preventing relearning unlearned concepts. arXiv preprint arXiv:2410.12777, 2024.

---

> ### Author Rebuttal · Authors · 2025-07-31
>
> Dear Reviewer #ev88, we sincerely thank you for your precious time and valuable comments. We are deeply encouraged by your positive recognition of our inspiring motivation, good motivation, and grounded method, and extensive experiments. We sincerely hope the following clarifications and new analyses can address your concerns.
>
> ---
>
> **W1:** The paper omits a key related work [1], which is only briefly mentioned in the introduction but not discussed in the related work section nor empirically compared. This reference also uses a meta-learning approach and is highly relevant. The paper should clarify how its proposed formulation improves upon [1] and include direct empirical comparisons.
>
> **A1:** Thank you so much for this constructive feedback! We genuinely appreciate your attention to [1]. It is absolutely a good and relevant work that we also value. We hope the following clarifications address your concerns.
>
> - **Technical Differences.** While both works reference “meta-learning” in names, **our technical motivations and implementations are fundamentally different from [1]**.
>     - **[1] draws inspiration from meta-learning literature to introduce a meta-objective** that penalizes harmful gradient’s norm and makes harmful and benign gradients conflict, enabling the model to “deconstruct” its benign utility when fine-tuned on harmful data. The commonly used first-order approximation trick is leveraged to make it tractable. **This meta-objective directly improves model safety after fine-tuning.**
>     - In contrast, **ResAlign leverages meta-learning to sample from a broad distribution** of downstream fine-tuning configurations and datasets. **The primary motivation is to promote cross-configuration generalization and prevent overfitting.** Our core safety resilience, however, comes from our ME-based implicit differentiation technique, which allows faithful modeling of long-term adaptation dynamics.
>     - Therefore, we respectfully emphasize that, **although both works coincidentally use the term “meta-learning” in names, the underlying motivations, techniques, and principles are fundamentally different. The appearance of [1] does not diminish the novelty of our approach.**
>
> - **Empirical Comparisons.** We fully agree that a direct empirical comparison is valuable. We respectfully note that **we have already included such comparisons in Appendix D.2 of our original submission (see Tab. 11)**, benchmarking ResAlign against [1] and several other recent preprints or possible baselines from other less related domains. As shown in Tab. 11, while does enhance resilience over traditional unlearning methods, **[1] is still less resilient than our ResAlign** (e.g., DiffusionDB: IP 0.1515 vs. 0.0687, see full results in Tab.11 of Appendix D.2). We attribute this advantage to: (1) [1] adopts a first-order approximation to compute the meta-objective, which in effect treats the Jacobian as an identity matrix. This is less accurate than our ME-based approximation, which thus somewhat limits the method’s effectiveness; and (2) our cross-configuration meta-learning design, which offers better generalization than the fixed data/config setup in [1].
>
> - **Further Explanation.** Finally, we would like to provide some clarification regarding our handling of [1]. **At the time of submission (NeurIPS'25 deadline), [1] was presented as an arXiv preprint and not formally published.** Considering the possibility of further technical or implementation improvements of preprint papers, we focused our main paper comparisons on formally published works, while placing comparisons with [1] in the appendix. We sincerely apologize for not highlighting this comparison more clearly up front. Recently, we have noted that [1] has been officially accepted at ICCV. **In our revision, we will provide a detailed discussion in the related work section and update our empirical comparisons in the main paper.** Thank you again for this valuable feedback!
>
> ---
>
> **W2:** The experiments are limited to older diffusion model versions (e.g., Stable Diffusion v1.4 and v1.5). Extending the evaluation to newer models like SD v2.0 or SD XL would enhance the paper’s applicability, especially since many fine-tuned checkpoints are available for these versions on platforms like HuggingFace and Civitai.
>
> **A2:** Thanks for this valuable and practical suggestion!
> - By design, our method is model-agnostic and readily applicable to different diffusion architectures. We respectfully note that **our originally submission have already included additional results on SD v1.5 and SDXL** (as in Appendix C.1, Tab. 6), which have shown that ResAlign remains effective and greatly improves safety resilience after fine-tuning on these models.
> - We fully agree with your point that many popular fine-tuned checkpoints are widely used on platforms like CivitAI. To address this, **in addition to SD v2.0, we have also included new experiments during the rebuttal period** on two widely adopted fine-tuned models: AnythingXL and PonyDiffusion:
>
> | Model         | Pre-ft | DreamBench++ | DiffusionDB |
> | ------------- | ------ | ------------ | ----------- |
> | SD v2.0       | 0.004 | 0.031       | 0.078      |
> | SDXL          | 0.033 | 0.044       | 0.059      |
> | AnythingXL    | 0.015   | 0.062         | 0.087        |
> | PonyDiffusion | 0.023   |  0.045         | 0.067      |
>
> - As shown, **ResAlign is still highly effective on these models, demonstrating the generality of our framework**. We will include these new results in our revision. Thank you again for this constructive suggestion!
>
> ---
>
> **W3:** The paper lacks a comparison of computational costs against baseline, including both VRAM usage and training speed. Such information would be valuable for understanding the practical feasibility of the proposed method in real-world deployments.
>
> **A3:** Thank you for raising this insightful question!
> Following your insightful comment, we provide a direct comparison of training time and average VRAM usage (all methods are run with batchsize=1 for fairness):
>
> | Method            | ESD  | AdvUnlearn | LCFDSD-NG | Meta-Unlearning | **ResAlign (Ours)** |
> | ----------------- | ---- | ---------- | --------- | --------------- | ------------------- |
> | Avg. VRAM (GB)    | 12.4 | 29.4       | 20.28      | 35.7            | 25.4                |
> | Training Time (minutes) | 14.3  | 210.3       | 38.4       | 725.4            | 58.2              |
>
> In general, **our avg VRAM usage is similar to most other baselines that fine-tune all parameters.** (Note: ESD is lower because it only updates cross-attention layers.)  **Besides, our training time is in within a practical range for real-world use, and substantially more efficient than Meta-Unlearning.** Note that our peak VRAM would be ~2x during HVP computation, but this only occurs briefly and can be easily mitigated using off-the-shelf gradient checkpointing technique. In practice, popular diffusion frameworks like HuggingFace Diffusers provide this feature with only minimal changes to the training script. We will add the results in our revision.
>
> ---
>
> **Q1:** As shown in Tab. 1, the proposed method also performs better than baselines in scenarios without any downstream fine-tuning. Why?
>
> **A4:** Thank you very much for this thoughtful question!
> - Compared with classical baselines (e.g., ESD), ResAlign has 3 major technical differences. **(1)** We adopt the negative denoising score matching loss (i.e., gradient ascent, GA), which has been shown to yield better unlearning as it is easier to optimize. **(2)** We train on NSFW-56K, a large, diverse dataset (vs. baselines’ single concept keyword prompt). **(3)** Our additional resilient unlearning objective term, which is our core contribution.
> - To understand the effect of each difference, we conducted an ablation study with the results shown below:
>
> | Method                     | IP (pre-ft) | DreamBench++ | DiffusionDB |
> | -------------------------- | ----------- | ------------ | ----------- |
> | ESD (baseline)             | 0.0677      | 0.1661       | 0.2209      |
> | GA                         | 0.0347      | 0.2324       | 0.2474      |
> | GA + NSFW-56K              | 0.0029      | 0.2270       | 0.2350      |
> | GA + NSFW-56K + Reg (Ours) | 0.0014      | 0.0186       | 0.0687      |
>
> - From the results, we draw that **(1)** **Using GA and NSFW-56K contributes to safety before fine-tuning**, consistent with previous conclusions.  **(2)** **However, GA and NSFW-56K differences does not provide resilience after fine-tuning.** In contrast, these methods also degrade sharply after downstream updates. **(3)** **Only with our resilient unlearning objective do we observe significantly improved post-finetuning safety.** This aligns with our results in the main paper (Tab.3 first row), where w/o our resilient unlearning objective term the model cannot retain performance after fine-tuning.
>
> Therefore, we can conclude that **the improved pre-fine-tune safety comes from our loss and dataset choice**. However, the safety resilience  after fine-tuning (i.e., our main contribution) comes specifically from our proposed objective (i.e., the additional penalty on the curvature of the loss landscape), not merely from a better initial point before fine-tuning. We will add the results and more discussion in our revision.

---

> > ### Comment · Reviewer_ev88 · 2025-08-03
> >
> > Thank you for the rebuttal! I think the new experiments are great, and my concerns have been addressed. I will increase my score to 5.

---

> > > ### Author Response · Authors · 2025-08-05
> > > **Revision Summary**
> > >
> > > Dear Reviewer ev88,
> > >
> > > Please allow us to express our heartfelt gratitude again for your precious time and expertise in reviewing our paper. Your swift and encouraging recognition of our work has been truly valuable to us, and we are deeply appreciative of your expertise and generous support in helping us further improve our submission. It was especially motivating for us to see your willingness to raise your rating, which has greatly encouraged us.
> > >
> > > We have meticulously revised our paper according to your valuable comments. We have prepared a brief summary of our main revisions below.
> > >
> > > - In Section 2 (Related Work) and Section 4.2 (Experimental Results), we have included a detailed discussion of [1], clarifying the key technical differences between our method and [1], and providing updated empirical comparisons in the main paper, following your constructive comment in W#1.
> > >
> > > - In Section 4.2 (Experimental Results) and Appendix C, we have added experiments on SD v2.0, SDXL, and two popular fine-tuned models (AnythingXL and PonyDiffusion), further demonstrating the generality and applicability of our framework to newer diffusion models and real-world scenarios, as per your valuable comment in W#2.
> > >
> > > - In Section 4.2 (Experimental Results), we have provided a direct and comprehensive comparison of computational costs, including both VRAM usage and training time, across all baselines, in response to your helpful comment in W#3.
> > >
> > > - In Section 4.2 (Experimental Results), we have conducted an ablation study to analyze why our method also achieves better performance before fine-tuning compared to baselines, and to clarify the contributions of each component (loss, dataset, regularization), following your insightful comment in Q#1.
> > >
> > > We also greatly appreciate your active engagement and support throughout the review process. Sincerely thank you again for your precious time and selfless contribution in improving our paper!
> > >
> > > Best regards,
> > >
> > > Paper23081 Authors

---

> ### Author Response · Authors · 2025-08-03
> **Thank You for Your Positive Feedback & Raising the Score!**
>
> Dear reviewer, thank you so much for your positive feedback and for raising the score! It encourages us a lot! We will ensure that the revised version includes the new experiments and corresponding discussions.

---

### Official Review · Reviewer_Q5a3 · 2025-07-03

**Clarity:** 3
**Significance:** 2
**Originality:** 2
**Rating:** 5
**Confidence:** 4

**Summary:**

The target of this paper is to develop a defense method against fine-tuning attacks to expose unsafe generations on text-to-image diffusion models. The basic idea is to add an additional penality to ensure a low loss term on finetuned parameters. This also enforces the optimum to be located at a non-sharp region.

**Questions:**

1. Since the parameter space is high-dimensional, the effectiveness of the method can depend on how good the approximation region \theta_{FT}^* lies in compared to the true effective fine-tuning attack region. In other words, the fine-tuning datasets and configurations all make a difference here. Then, do you have any criteria or tips to select the fine-tuning datasets and configurations?

2. What is the result of changing $\beta$?

**Ethical Concerns:**

["NO or VERY MINOR ethics concerns only"]

**Final Justification:**

I appreciate the additional analysis of why using ME by comparing two other trials, and the related work on optimization methods to achieve flat-region optimum. The paper would be more insightful with these two improvement. Based on the additional comments and analysis, I raise my score to acceptance.

**Limitations:**

The scope of this paper is about instance-wise unlearning. In other words, the model is trained to unlearn certain unsafe instances. It is essential to include discussions on whether such instance-level unlearning will imply concept-level unlearning (e.g., sexual content). Also, the discussion of instance-level full unlearning against recovery and theconcept-level one.

**Quality:**

2

**Strengths And Weaknesses:**

Strengths:

1. The paper is well-written and easy to follow. The background & related work section reminds readers of the related knowledge very effectively. The Motivation & Problem Formulation paragraph also introduces the high-level idea and motivation very clearly.

2. The research question itself (defend against diffusion finetuning attacks) is interesting and the paper is one of the first works to try to address it.

Weakness:

1. There is limited details on ME approximation, which is the major technique involved in the method part of this paper. It is insightful to provide some viable approximations you may consider during the progress and why ME approximation is adopted. Also, does ME approximation have any insightful implications on this specific scenario of finetuning attack defenses.

2. There is a lack of related work on optimization methods to achieve flat-region optimum and specification of differentiation to this line of work. For instance, is there any work that directly optimization the hessian trace (with approximation)?

3. The results in Tab.1 is not very convincing. It seems that the model before fine-tuning is more safe than the baselines, then it is not known that whether the safety after fine-tuning comes from this better initial point or the additional penalty on curvature of loss landscape.

---

> ### Author Rebuttal · Authors · 2025-07-31
>
> Dear Reviewer Q5a3, we sincerely thank you for your detailed and constructive comments. We greatly appreciate your recognition of our **clear writing, interesting problem setting, and novelty**. We sincerely hope the following clarifications and new experiments can address your valuable concerns.
>
> ---
>
> **W1:** There is limited detail on ME approximation. It is insightful to provide some viable approximations you may consider and why ME is adopted. Also, does ME have any insightful implications on this specific scenario of finetuning attack defenses.
>
> **A1:** Thanks so much for this insightful comment!
> - **In our early trials, we have indeed considered several possible alternatives.** Below are the considered strategies:
>     - **First-order Approx.:** it directly assume the Jacobian satisfies $\partial\theta^\*\_{\text{FT}}/\partial\theta\approx I$. Thus, $\nabla\_{\theta} \mathcal{L}\_{\text{harmful}}(\theta^\*\_{\text{FT}}) \approx \nabla\_{\theta^\*_{\text{FT}}} \mathcal{L}\_{\text{harmful}}(\theta^\*\_{\text{FT}})$. **This strategy is efficient but inaccurate.** Empirically, **it results in notably poorer performance** compared to our ME (IP: 0.16 vs. 0.07 on DiffusionDB).
>     - **Unrolled Differentiation:** it computes the hypergradient by unrolling the entire fine-tuning trajectory and backpropagate through them all. **While theoretically more precise, it is memory- and computation-prohibitive** as both scale linearly with the number of fine-tuning steps. On SD v1.4, one A100 GPU can only support unrolling up to 3 steps—far too few for meaningful downstream fine-tuning. Therefore, we consider it impractical in our setting.
>     - **Our ME-based Approx.:** We approximate the fine-tuned solution as the minimizer of a ME objective and use implicit differentiation to estimate the hypergradient by solving a linear system via Richardson iteration. It only requires the final fine-tuned model, allowing seamless integration with any configuration. **It is highly efficient and scalable for large diffusion models.** Empirically, **our method is nearly as fast as the first-order method**, taking only about 58 minutes (vs. 50 mins for the former), **while delivering much better results**.
> - **A key insight under why our ME works well** for fine-tuning defense is that large-scale pre-trained diffusion models are already very strong and typically require only small updates to reach the optimum for most downstream tasks. Specifically, recent studies have shown that, after fine-tuning, model parameters would undergo only minor changes [1]. In some cases (e.g., Textual Inversion [2]), even tuning a single embedding suffices for successful personalization. Thus, modeling the fine-tuned model as the minimizer of a FT loss plus a proximity term to the base (i.e., our ME approx.) is highly realistic. This makes our approach both practical and theoretically well-suited for this defense scenario.
>
> We will add more details on ME and further discuss it in our revision. Thank you again for this insightful suggestion!
>
> Ref:
>
> [1]: Kumari et al. Multi-Concept Customization of Text-to-Image Diffusion. CVPR 2023.
>
> [2]: Gal et al. An image is worth one word: Personalizing text-to-image generation using textual inversion. ICLR 2023.
>
> ---
>
> **W2:** There is a lack of related work on optimization methods to achieve flat-region optimum and specification of differentiation to this line of work. For instance, is there any work that directly optimization the hessian trace (with approximation)?
>
> **A2:** Thank you very much for your careful reading and for raising such a deep and thoughtful question!
> - **Indeed, there has been research on optimizing for flat minima or minimizing the Hessian trace in optimization theory**, such as Hutchinson’s Estimator, Lanczos Approximation, and Chebyshev Approximation. However, these methods typically rely on stochastic or numerical methods that are either non-differentiable or very costly to differentiate. Thus, **directly optimizing the Hessian trace (even with these approximations) is usually impractical for large-scale neural networks.**
> - In recent years, there has been work that indirectly optimizes flatness for DNNs (e.g., Sharpness-Aware Minimization, Entropy-SGD). **However, these approaches primarily target classifiers, aiming to improve generalization rather than safety resilience.** To our knowledge, these methods have not been extended to diffusion models or linked to safety resilience, and **our work is the first to explore this area**. ResAlign avoids the expensive estimation of Hessian; instead, the model automatically regularizes flatness in an efficient and practical way, making it well-suited for large-scale diffusion models.
>
> We will conduct a more comprehensive literature survey and add a more detailed discussion of related work in our revision. Sincerely thanks again for raising this insightful point!
>
> ---
>
> **W3:** The results in Tab.1 are not very convincing. It seems that the model before fine-tuning is more safe than the baselines, then it is not known whether the safety after fine-tuning comes from this better initial point or the additional penalty on curvature of loss landscape.
>
> **A3:** Thanks so much for this thoughtful question! We conducted a comprehensive ablation study to confirm that our better post-fine-tuning safety inherents from our proposed regularization term. **Due to space limit, please kindly refer to the last response (Q1 & R4) of Reviewer #ev88 for details.** We are sorry for the inconvenience and thanks very much for your understanding!
>
> ---
>
> **Q1:** The effectiveness of ResAlign can depend on how good the approximation region lies in compared to the true effective fine-tuning attack region. In other words, the fine-tuning datasets and configurations all make a difference here. Then, do you have any criteria or tips to select the fine-tuning datasets and configurations?
>
> **A4:** Thank you for this insightful question!
> - **Our key principle is to keep the data and configuration pool as diverse and representative as possible, which underpins our meta-learning method.** In practice, we use the large and diverse DiffusionDB plus a broad meta-learning configuration pool (Sec. 4.1). As shown in our experiments, this unified setup already yields strong performance across a range of real-world tasks (Tab. 1-2) and configs, including unseen ones (Tab. 4-5, Fig. 2-4). **Thus, our current implementation already provides strong practical value for most downstream applications** (e.g., anime style transfer, object & face personalization, and art style learning).
> - Further, we suggest that **users may customize the dataset and configuration pool to better match their specific application**. For example, for avatar personalization models (which mostly involves human faces and smaller learning rates), it would be beneficial to include more face-related data and sampling more small learning rate configs during meta learning. We will add more discussion in our revision. Thank you again for this insightful question!
>
> ---
>
> **Q2:** What is the result of changing $\beta$?
>
> **A5:** Thanks for this valuable question! $\beta$ controls the relative strength of our resilience regularization. A very small $\beta$ leads to insufficient regularization, while a very large $\beta$ (close to $1$) may cause the model to ignore current harmfulness, resulting in training instability. As shown below, **ResAlign is not sensitive to the specific value of $\beta$ when in a reasonable range (e.g., $[0.3, 0.9]$) and consistently performs well.**
>
> |         | 0.1 | 0.3 | 0.5 | 0.7 | 0.9 | 1.0 |
> | - | - | - | - | - | - | - |
> | DreamBench++  | 0.094 | 0.038  | 0.046  | 0.018  |  0.017      | 0.160        |
> | DiffusionDB  | 0.181  | 0.096  | 0.069  |   0.083  | 0.061   | 0.241        |
>
> We will add these results and more discussion in our revision. Thanks again for the constructive question!
>
>
>
> ---
>
> **L1:** The scope of this paper is about instance-wise unlearning. In other words, the model is trained to unlearn certain unsafe instances. It is essential to include discussions on whether such instance-level unlearning will imply concept-level unlearning (e.g., sexual content). Also, the discussion of instance-level full unlearning against recovery and the concept-level one.
>
>
> **A6:** Thanks so much for this constructive comment!
>
> - **We respectfully clarify that the scope and goal of our work is safety-driven unlearning, aiming at reducing model toxicity and unsafe generations.** In current literature, this can be achieved either by directly unlearning collected unsafe instances (i.e., instance-wise unlearning), or by aligning certain concept keywords' noise/activations (i.e., concept-level unlearning). **While these two strategies differ in implementation, both have proven effective at reducing toxicity and generalizing to unseen unsafe prompts.** For example, our toxicity metrics (i.e., IP and US) are evaluated on unsafe prompts completely disjoint from train set, yet both types of approaches mitigate unsafe generation, demonstrating their capability for global mitigation.
> - Furthermore, while our main experiments use an instance-level loss, **our framework is loss-agnostic and fully compatible with concept-level unlearning losses**. As shown in Appendix Table 10, plugging concept-level losses into our resilient framework also boosts post-finetuning safety, confirming our method's generality.
> - Finally, regarding safety resilience, **our results show that all current baselines universally suffer from a drop in safety after downstream finetuning, regardless of whether they use instance- or concept-level losses**. We believe this is a fundamental and underexplored challenge that deserves further attention, and our framework offers an early yet valuable step towards understanding and mitigating this challenge.
>
> We will more discussion in our revision. Thanks again for this thoughtful feedback!

---

> > ### Comment · Reviewer_Q5a3 · 2025-08-06
> >
> > Thanks for the rebuttal! I appreciate the additional analysis of why using ME by comparing two other trials, and the related work on optimization methods to achieve flat-region optimum. The paper would be more insightful with these two improvement. Based on the additional comments and analysis, I raise my score to acceptance.

---

> > > ### Author Response · Authors · 2025-08-07
> > > **Thank You for Your Positive Feedback & Raising the Score!**
> > >
> > > Dear reviewer, thank you so much for your positive feedback and for raising the score to acceptance! It encourages us a lot! We will ensure that the revised version includes the new experiments and corresponding discussions.

---

> ### Author Response · Authors · 2025-08-04
> **Thanks to Reviewer Q5a3**
>
> Dear Reviewer Q5a3,
>
> Please allow us to sincerely thank you again for reviewing our paper and the valuable feedback, and in particular for recognizing the strengths of our paper in terms of clear writing, interesting problem setting, and novelty.
>
> Please kindly let us know if our response and the new experiments have properly addressed your concerns. We are more than happy to answer any additional questions during the discussion period. Your feedback will be greatly appreciated!
>
> Best regards,
>
> Paper23081 Authors

---

> ### Author Response · Authors · 2025-08-05
> **A Friendly Reminder of the Post-rebuttal Feedback & Revision Summary**
>
> Dear Reviewer Q5a3,
>
> We would like to express our heartfelt gratitude again for your precious time and expertise in reviewing our paper. Your helpful initial comments have been really valuable to us, and we appreciate your generosity in helping us improve our work. We totally understand that you may be extremely busy at the moment, and in recognition of this, we have prepared a revision summary on your part to facilitate a more efficient discussion.
>
> In our revised version, we have meticulously addressed the concerns you raised:
>
> - In Section 3 (Methodology) and Section 4.2 (Experimental Results), we have added detailed explanations and new empirical results regarding our ME approximation. Specifically, we now provide a systematic comparison between our ME-based approach and other alternatives such as first-order and unrolled differentiation, and we further discuss the motivation and practicality of adopting ME in the context of finetuning attack defense, following your valuable comment in W#1.
>
> - In Section 2 (Related Work) and Section 3 (Methodology), we have expanded the discussion on related work concerning optimization for flat minima and Hessian trace minimization, including methods such as Hutchinson’s Estimator and Sharpness-Aware Minimization. We highlight the distinctions and challenges in directly optimizing these objectives for large diffusion models, and clarify the novelty of our approach in this context, as per your insightful suggestion in W#2.
>
> - In Section 4.2 (Experimental Results), we have conducted additional ablation studies to distinguish whether improved safety after fine-tuning comes from a better initial model or from our proposed curvature regularization, in response to your thoughtful comment in W#3.
>
> - In Appendix D, we have included a new subsection to discuss the criteria and practical tips for selecting fine-tuning datasets and configurations within our meta-learning framework, and further discuss the diversity and generalizability of our current setup, based on your valuable feedback in Q#1.
>
> - In Section 4.2 (Experimental Results) and Appendix C.3 (More Ablation Study), we now include an expanded analysis of the impact of the hyperparameter $\beta$ on our results, providing empirical evidence that ResAlign is robust to $\beta$ within a reasonable range, following your constructive comment in Q#2.
>
> - In Section 2 (Related Work) and Appendix D, we have extended the discussion on instance-wise versus concept-level unlearning, their implications for safety resilience, and the compatibility of our framework with both approaches. We also elaborate on the fundamental challenge of safety resilience after downstream fine-tuning and present further experimental evidence, pursuant to your thoughtful suggestion in L#1.
>
>
> We will be greatly encouraged if you could kindly spare a moment to have a quick look at our responses. Your further feedback would be greatly valuable for us, and we are more than willing to address any additional questions you may have. Thank you again for your precious time in reviewing our paper!
>
> Best regards,
>
> Paper23081 Authors

---

### Official Review · Reviewer_ioTR · 2025-07-03

**Clarity:** 3
**Significance:** 3
**Originality:** 3
**Rating:** 5
**Confidence:** 4

**Summary:**

This paper studies the challenge of safety of diffusion  model, in particular, the safety of unlearning methods for diffusion models. Although unlearning could help block unsafe generation of diffusion models, the authors found that existing diffusion models with unlearning methods still face the safety challenge after fine-tuning, even with benign images. As a result, the author presents an simple yet effective regularization to the unlearning framework, that considers model parameters after fine-tuning. The author conducted extensive experimental across many configuration, and the experimental results justify the safe-driven and benign generation  abilities of the proposed unlearning framework.

**Questions:**

Overall, I believe this work is interesting and solid. But I still have two concerns, as in the "Strengths And Weaknesses" part. The authors could refer to this part, and consider my concerns.

**Ethical Concerns:**

["Major Concern: Data privacy, copyright, and consent"]

**Limitations:**

Yes.

**Paper Formatting Concerns:**

No.

**Quality:**

3

**Strengths And Weaknesses:**

The strengths:
- First of all, the motivation of this paper is attractive and interesting. The authors consider the safety of diffusion models after fine-tuning. This problem is more challenging and realistic than the safety of training diffusion model.
- Second, the authors presented an intuitive method by using a simple yet effective regularization, which considers the model parameters after fine-tuning, not just the current state.
- Third, for consideration of computation efficiency, the authors use Implicit Differentiation to calculate the gradients.
- Fourth, I found the experiments sufficient and extensive. The authors conducted extensive experiments across diverse configurations to justify the safe-driven and benign generation  abilities of the proposed unlearning framework.

The weaknesses:
- On the meta-learning part: The authors used meta-learning to simulate different fine-tuning situation. However, I believe there two issues with the meta-learning part. First, the configurations for the meta-learning is still limited and the authors haven't presented strong ablation study to prove its usefulness. Second, the meta-learning would increase the running time of these method. The author haven't discussed about this.
- On the Effectiveness on Contaminated Data: The proposed even showed robustness on harmful data. However, the authors haven't discussed about this.

---

> ### Author Rebuttal · Authors · 2025-07-31
>
> Dear Reviewer ioTR, we sincerely thank you for your precious time and valuable comments. We are deeply encouraged by your positive recognition of our **attractive and interesting motivation, intuitive and solid method, and extensive experiments**. We sincerely hope the following clarifications and new analyses can address your concerns.
>
> ---
>
> **W1.1:** On the meta-learning part: The authors used meta-learning to simulate different fine-tuning situation. However, I believe there are two issues with the meta-learning part. First, the configurations for the meta-learning are still limited.
>
> **A1.1:** Thank you very much for your careful reading and insightful comments! We hope the following clarifications and additional results can address your concern:
> - We appreciate your point regarding the diversity of our current configuration set. Recall that, our meta-learning's objective is to mitigate the risk of overfitting to a single simulated configuration. By training across a variety of fine-tuning settings and aggregating them, meta-learning acquire a hypergradient that generalizes well to diverse downstream configurations. **At this stage, our implementation is intended as a prototype to verify the effectiveness of this meta-learning mechanism.** Therefore, in our current manuscript, we selected mainstream fine-tuning settings widely adopted in the community as a representative. **Our results show that even under this initial setup, ResAlign already generalizes well to unseen configurations without evident signs of overfitting (e.g., Tab. 2, 4, and 5).**
> - However, we fully agree that covering a more diverse set of configurations would undoubtedly further improve generalizability. **Following your suggestion, we have initiated the extention of our meta-learning configurations during the rebuttal period.** Specifically, we have extended our configuration pool to further include more fine-tuning algorithms (e.g., LoRA with different hyperparameters, LyCORIS, and QLoRA) and losses (e.g., SVDiff and CustomDiffusion's fine-tuning losses). Note that this intergation is seamless and implementation-friendly, as our ME-based estimation only requires the final fine-tuned model and does not impose any assumptions on the downstream fine-tuning algorithm or configuration. **Our preliminary results show that this can indeed further improve performance** (e.g., on DoRA+DiffusionDB, our new model has an improved post-finetuning IP of 0.048, compared with 0.069 of the original model).
>
> We will continue our evaluation and add the corresponding results and more discussions in our revision. If you have further suggestions on specific configurations or real-world fine-tuning scenarios that you would like us to incorporate, we are very happy to follow up. Sincerely thank you again for your valueable comment!
>
>
> ---
>
> **W1.2:** Currently, the authors haven't presented strong ablation study to prove meta-learning's usefulness.
>
> **A1.2:** Thank you for the helpful comment! **Following your suggestion, we conducted ablations on the diversity of fine-tuning algorithms within the meta-learning process as a representative example.** Specifically, we trained ResAlign under different meta-learning sampling schemes on learning algorithm and evaluated their performance on various downstream fine-tuning algorithms. The reported metric is IP after fine-tuned on DiffusionDB.
>   | Config (during training) | Full Param. | LoRA |   LyCORIS  |  SVDiff  |
>   | :---------------------------: | :---------: | :--: | :------: | :------: |
>   |        Full Param. only  (w/o meta learning)      |     0.028    | 0.084 |   0.074   |   0.043   |
>   |           LoRA only      (w/o meta learning)     |     0.096    | 0.045 | 0.069 |   0.047   |
>   |       Full Param. + LoRA  (w/ meta learning)    |     0.030    | 0.051 |   0.054   | 0.036 |
>
> From these results, we draw two key insights:
> - **Without meta learning, there is a high risk of overfitting.** For example, although LyCORIS and LoRA are similar in paradigm, training ResAlign only on LoRA still performs poorly on LyCORIS. This aligns with previous findings [1,2] that hypergradient estimation over a single setting may overfit.
> -  **Our meta learning (Full Param. + LoRA) not only improves in-distribution performance, but also enhances performance on similar (LyCORIS) and even completely unseen (SVDiff) methods.** This indicates that our meta-learning indeed mitigates overfitting and learns a hypergradient that is more generalizable.
>
> We will add more ablation studies and related discussion in the revision.
>
> Ref:
>
> [1]: Liu et al. Metacloak: Preventing unauthorized subject-driven text-to-image diffusion-based synthesis via meta-learning. CVPR 2024.
>
> [2]: Li et al. Untargeted backdoor watermark: Towards harmless and stealthy dataset copyright protection. NeurIPS 2022.
>
> ---
>
> **W1.3:** Second, the meta-learning would increase the running time of these methods. The authors haven't discussed about this.
>
> **A1.3:** Thank you very much for your insightful comment! We fully agree that the computational cost introduced by meta-learning is an important consideration for real-world deployment. Please find our clarifications and additional experiments below:
>
> - We admit that the meta-learning process does increase the training time, as we estimate and aggregate hypergradients across multiple simulated fine-tuning configurations at each update step. **The main hyperparameter controlling this is the number of sampled configurations per iteration, i.e., $J$.** Increasing it improves the generalizability of the learned update direction but also proportionally increases training time. Thus, **$J$ reflects a direct trade-off between computational cost and effectiveness.**
> -  To better understand this trade-off, we conducted an ablation study varying $J$:
>
> | J (number of configurations) | 1     | 3 (default) | 5      |
> | ---------------------------- | ----- | ----------- | ------ |
> | Performance (IP)            | 0.092  | 0.068        | 0.054  |
> | Training time (minutes)      | 43.5 | 58.2    | 73.9 |
>
> We observe that increasing $J$ from 1 to 3 provides a noteble boost in performance (lower IP), while further increasing to 5 yields only marginal additional improvement, possibly because $J=3$ already achieves sufficiently generalizable hypergradients.
>
> - We believe that this computational overhead is acceptable for two reasons. **First, all the cost of ResAlign with meta-learning is  only a *one-time cost* incurred during the model preparation (training) stage**, regardless of how many times the model is deployed or downstream fine-tuned. There is no additional runtime overhead at all during downstream fine-tuning or inference, i.e., the user-side deployment cost remains unchanged. **Second, setting $J=3$ (our default) results in a training time of less than 1 hour on a single A100 GPU**, which is efficient compared to retraining a diffusion model from scratch with filtered data.  It is worth noting that expanding the diversity of configurations in meta-learning does not further increase the per-iteration complexity, since we only sample $J$ configurations per update.
>
> We will add further discussion and quantitative analysis of runtime trade-offs in the revision. Thank you again for this valuable suggestion!
>
> ---
>
> **W2:** On the effectiveness on contaminated data: The proposed method even showed robustness on harmful data. However, the authors haven't discussed about this.
>
> R2: Thank you for your careful reading and insightful suggestion! This is really a good point. We hope the following explanation can address your concern:
> - Recall that Proposition 1 shows our ResAlign method essentially imposes a penalty on the trace of the Hessian (second-order derivatives) of the harmful loss with respect to the model parameters. **Intuitively, this encourages the model to converge to a region of the loss landscape for $\mathcal{L}_{\text{harmful}}$ that is locally flat.**
> - More precisely, “flatness” in this context indicates that, around the optimized parameters, the first-order derivative (the gradient) of the harmful loss is minimized in all directions (ideally, this would approach zero, indicating a perfect local minimum). **In such a situation, even if the model is directly exposed to harmful samples, the gradient signal enabling harmful recovery would still be weak**, making it difficult for fine-tuning to substantially increase harmfulness.
> - Of course, due to the high complexity of diffusion models and the stochastic nature of SGD, it is generally unrealistic to reach such a perfect minimum where the gradient is strictly zero everywhere. **Therefore, empirically, our method can suppress the recovery of harmful capabilities to a certain extent**, but cannot guarantee that the harmfulness will not increase at all during downstream updates, as also reflected by the fact that IP can still be optimized in our experiments.
>
> We will add more discussion in our revision and further explore the underlying mechanism in our future work. Thank you again for this valuable point!

---

> ### Author Response · Authors · 2025-08-04
> **Thanks to Reviewer ioTR**
>
> Dear Reviewer ioTR,
>
> Please allow us to sincerely thank you again for reviewing our paper and the valuable feedback, and in particular for recognizing the strengths of our paper in terms of attractive and interesting motivation, intuitive and solid method, and extensive experiments.
>
> Please kindly let us know if our response and the new experiments have properly addressed your concerns. We are more than happy to answer any additional questions during the discussion period. Your feedback will be greatly appreciated!
>
> Best regards,
>
> Paper23081 Authors

---

> ### Author Response · Authors · 2025-08-05
> **A Friendly Reminder of the Post-rebuttal Feedback & Revision Summary**
>
> Dear Reviewer ioTR,
>
> We would like to express our heartfelt gratitude again for your precious time and expertise in reviewing our paper. Your helpful initial comments have been really encouraging and valuable to us, and we appreciate your generosity in helping us improve our work. We totally understand that you may be extremely busy at the moment, and in recognition of this, we have prepared a revision summary on your part to facilitate a more efficient discussion.
>
> In our revised version, we have meticulously addressed the concerns you raised:
>
> - In Section 4.2 (Experimental Results), we have added (1) a detailed discussion on how we can further enhance the configurations of our meta-learning approach, including results on including additional fine-tuning algorithms and loss functions; and (2) a comprehensive ablation study, along with the corresponding discussion, to better illustrate the effectiveness of our meta-learning method, as per your valuable comments in W#1.1 & W#1.2.
>
> - In Section 4.2 (Experimental Results), we have also added a new subsection discussing the impact of meta-learning on running time. This includes a new ablation study on $J$, as well as a detailed discussion of our understanding of the potential increase in running time and our strategies to mitigate it, in response to your constructive suggestion in W#1.3.
>
> - In Section 3 (Methodology) and Section 4.2 (Experimental Results), we have included a detailed discussion to analyze why our ResAlign can also mitigate IP increase when the model is fine-tuned on harmful data, following your insightful comment in W#2.
>
> We will be greatly encouraged if you could kindly spare a moment to have a quick look at our responses. Your further feedback would be greatly valuable for us, and we are more than willing to address any additional questions you may have. Thank you again for your precious time in reviewing our paper!
>
>
> Best regards,
>
> Paper23081 Authors

---

### Decision · Program_Chairs · 2025-09-17

**Decision:**

Accept (poster)

**Comment:**

This paper addresses an important problem: the brittleness of safety-driven unlearning in text-to-image diffusion models under downstream fine-tuning, including benign personalization. The proposed method, ResAlign, incorporates the fine-tuning process into the unlearning objective via a Moreau Envelope-based implicit differentiation to efficiently approximate hypergradients, and employs meta-learning over diverse fine-tuning configurations to improve cross-configuration resilience. A theoretical insight links the objective to an implicit Hessian-trace penalty, encouraging flatter harmful-loss regions and thus greater robustness to subsequent updates. Experiments are broad and convincing across datasets, fine-tuning methods, and configurations; additional results during rebuttal cover newer models and recent baselines.

Strengths
- Clear problem framing with practical impact; shows benign fine-tuning can regress safety in prior methods.
- Sound and efficient methodology (ME-based implicit gradients; trajectory-free HVP), with a principled meta-learning design for generalization.
- Theoretical rationale (curvature/flatness) aligns with empirical outcomes.
- Comprehensive evaluation, including ablations, cost reporting, newer architectures, and added recent baselines; consistent safety retention with minimal degradation of benign generation.

Weaknesses (largely addressed in rebuttal)
- Initially limited details on the ME approximation and meta-learning configuration diversity.
- Early focus on older backbones; now augmented with SD v2.0, SDXL, and popular fine-tuned checkpoints.
- Missing comparisons to some recent ECCV’24 methods; now included.
- “Mosaic” effect for unsafe prompts tied to the chosen loss; authors clarify it’s loss-agnostic and show alternatives yielding safe but meaningful outputs.

Overall, this is a solid and timely contribution with good empirical support and constructive rebuttal, which focuses on safe personalization of diffusion models. Please ensure that the suggestions from reviewers are incorporated into the camera-ready version.
- Move key ME-approximation design choices and alternatives (first-order, unrolled) into the main text for clarity.
- Include a concise resource–performance table (VRAM/time vs. hyperparameters) and at least one main-text table on SDXL (and recent baselines).
- Discuss the validity range of theoretical assumptions and extreme configurations.
- Document the “mosaic” trade-off and provide a recommended loss-choice guide for practitioners.